# Population Priors for Matrix Factorization

**Sohrab Salehi**                                          *sohrab.salehi@columbia.edu*
*Computational Oncology, Department of Epidemiology and Biostatistics, Memorial Sloan Kettering Cancer Center, New York, NY, USA*
*Irving Institute for Cancer Dynamics, Columbia University, New York, NY, USA*

**Achille Nazaret**                                          *aon2108@columbia.edu*
*Department of Computer Science, Columbia University, New York, NY, USA*

**Sohrab P Shah**                                          *shahs3@mskcc.org*
*Computational Oncology, Department of Epidemiology and Biostatistics, Memorial Sloan Kettering Cancer Center, New York, NY, USA*

**David M Blei**                                          *david.blei@columbia.edu*
*Department of Statistics, Columbia University, New York, NY, USA*
*Data Science Institute, Columbia University, New York, NY, USA*
*Department of Computer Science, Columbia University, New York, NY, USA*

**Reviewed on OpenReview:** *https://openreview.net/forum?id=AT9G5s1pOj*

## Abstract

We develop an empirical Bayes prior for probabilistic matrix factorization. Matrix factorization models each cell of a matrix with two latent variables, one associated with the cell's row and one associated with the cell's column. How to set the priors of these two latent variables? Drawing from empirical Bayes principles, we consider estimating the priors from data, to find those that best match the populations of row and column latent vectors. Thus we develop the twin population prior. We develop a variational inference algorithm to simultaneously learn the empirical priors and approximate the corresponding posterior. We evaluate this approach with both synthetic and real-world data on diverse applications: movie ratings, book ratings, single-cell gene expression data, and musical preferences. Without needing to tune Bayesian hyperparameters, we find that the twin population prior leads to high-quality predictions, outperforming manually tuned priors.

## 1 Introduction

This paper is about empirical Bayes methods for setting the priors in Bayesian matrix factorization (Mnih & Salakhutdinov, 2007; Gopalan et al., 2015). Matrix factorization models each cell of a matrix with two latent variables, one associated with its row and one associated with its column. Matrix factorization has found broad applications across many fields, including studying consumer behavior, understanding legislative patterns, assessing pharmaceutical impacts, and exploring social networks (Gopalan et al., 2015; Koren et al., 2009; Gerrish & Blei, 2011; Jamali & Ester, 2010).

Suppose $X_{i,j}$ is the observed entry in row $i$ and cell $j$, such as user $i$'s rating of movie $j$. As a hierarchical model, a matrix factorization generates the data from the following process:

$$\boldsymbol{U}_i \sim \pi_{\text{row}}(\boldsymbol{U}_i) \tag{1}$$
$$\boldsymbol{V}_j \sim \pi_{\text{col}}(\boldsymbol{V}_j) \tag{2}$$
$$X_{i,j} \sim P(X_{i,j} \mid \boldsymbol{U}_i, \boldsymbol{V}_j). \tag{3}$$

Here, $\boldsymbol{U}_i$ and $\boldsymbol{V}_j$ are per row and per column specific latent vectors, and $\pi_{\text{row}}$ and $\pi_{\text{col}}$ are the priors distributions.

This formulation encompasses many factorization models. In Gaussian matrix factorization (Mnih & Salakhutdinov, 2007), the priors are Gaussians and $X_{i,j}$ is drawn from a Gaussian with mean $\boldsymbol{U}_i^\top \boldsymbol{V}_j$ and variance $\sigma^2$. In Poisson matrix factorization (Canny, 2004; Dunson & Herring, 2005; Gopalan et al., 2015), the priors are over the positive reals and $X_{i,j}$ is drawn from a Poisson with rate $\boldsymbol{U}_i^\top \boldsymbol{V}_j$.

An observed matrix of data $\mathbf{X}$ then defines a posterior distribution $P(\boldsymbol{U}, \boldsymbol{V} \mid \boldsymbol{X})$ over the row variables and the column variables. The posterior can provide interpretations of the data and an avenue to form predictions about missing entries, for example, for a recommendation system.

The prior distributions on the row variables and column variables significantly impact the quality of the model posterior. How should we set them? Practitioners typically assume a simple parametric family for the priors, such as a Gaussian or a Gamma, and then find the prior hyperparameters best suited to the data, e.g., with cross-validation (Salakhutdinov & Mnih, 2008; Schmidt et al., 2009). This approach can be effective, but it is expensive and only allows for priors from a simple parametric family.

In this paper, we develop an empirical Bayes (EB) methodology for setting the priors (Robbins, 1992; Efron, 2012), learning them from data. The EB idea is to set the priors using the data, for example by finding the one that maximizes the marginal log-likelihood of the data. The EB idea is applicable to any prior distribution from which are repeatedly drawn multiple independent variables. For example, all the $\mathbf{U}_i$ are independently drawn from the same $\pi_{\text{row}}$ and the $\mathbf{V}_j$ are independently drawn from the same $\pi_{\text{col}}$. EB has found applications in applied sciences in diverse fields such as astronomy (Bovy et al., 2011), actuarial sciences (Bühlmann & Gisler, 2005), genomics (Smyth, 2005; Love et al., 2014), economics (Frost & Savarino, 1986; Angrist et al., 2017), and survey sampling (Rao & Molina, 2015). EB priors have been successfully employed in simple hierarchical models, such as in variational autoencoders (Kingma & Welling, 2013; Tomczak & Welling, 2018; Kim & Mnih, 2018).

Matrix factorization, however, provides a different type of application of EB. In matrix factorization, there are two priors to set, one for the row variables and one for the column variables, and the same data contains information about them, namely the observed data $\mathbf{X}_{i,j}$ contains information about both $\mathbf{U}_i$ and $\mathbf{V}_j$. Thus, we will find empirical Bayes priors for both row variables and column variables. The result is the *twin empirical Bayes prior* (TwinEB), a practical EB method for matrix factorization. Other methods for EB on matrix factorization include Wang & Stephens (2021) and da Silva et al. (2023); we discuss these in section 2.

Specifically, we model the two priors with mixture distributions, one for each prior. We use mixtures since they are a flexible family of distributions that can approximate a wide range of distributions (Titterington et al., 1985; Nguyen et al., 2020). We use variational inference (Blei et al., 2017; Wainwright & Jordan, 2008) to simultaneously estimate the priors and approximate the corresponding posterior. We verify the efficiency and the robustness of this approach with real-world data about recommendation systems and computational biology, and for both Poisson matrix factorization and Gaussian matrix factorization.

We summarize the contributions as follows:

1. We develop the *twin population prior*, an EB prior for Bayesian matrix factorization.

2. We derive a variational inference algorithm to approximate the matrix factors' posteriors and learn the twin population prior simultaneously.

3. We study the twin population prior on both synthetic and real data and with two types of factorization. The automatically learned EB prior performs as well as the best prior chosen retrospectively.

## 2 Related work

One approach to setting the hyperparameters of priors involves using hierarchical Bayesian models (Gelman, 2006). In this line of work, the prior's hyperparameters are treated as unknown and assigned a prior with hyperpriors to be determined. Gopalan et al. (2015) employs this method for introducing hierarchical Poisson factorization, and Levitin et al. (2019) applies it to gene signature discovery from scRNAseq data. Our work involves learnable priors (mixtures), which leads to a more flexible class of models while keeping a simpler model as no extra variables for the hyperparameters are introduced.

A fruitful direction of research for setting priors for matrix factorization has also been the use of empirical Bayes, a methodology by which one tries to find the prior that matches, in some sense, the population distribution of the data. In Rukat et al. (2017), the authors set a single and fixed values for the prior hyperparameter based on the expected value of the observed matrix. In Wang & Stephens (2021), the authors formulate prior elicitation for matrix factorization (MF) as steps in a variational expectation maximization algorithm, and in that context, find that it is equivalent to solving the EB normal means (EBNM) problem (Jiang & Zhang, 2009). Our proposed approach is closely related to Wang & Stephens (2021), but we propose to update both priors simultaneously which bypasses potentially costly numerical integration required in the EBNM step. A related line of research makes the connection to EBNM via random-matrix theory, focusing specifically on denoising principal components (Zhong et al., 2022). Closely related to TwinEB, da Silva et al. (2023) optimizes hyperparameters to match the prior predictive distribution to statistics computed from the data, and derives closed-form solutions for PMF and its hierarchical extensions. In Section 4, we compare our method to Wang & Stephens (2021) and da Silva et al. (2023). In Appendix E, we give a more detailed review of these methods.

Wang et al. (2024) uses auxiliary information to improve inference in matrix factorization. They assume the existence of an auxiliary matrix $G$, comprising per user extra features. Their method is closest to that of Wang & Stephens (2021), except that they define the per row latent variables as a function of the auxiliary matrix $G$. They use an ensemble of regression trees to model the functional dependence of the row-wise latents on the auxiliary variables. They use a Gaussian likelihood, with a mean as a linear function of the user and item representations.

Van Linh et al. (2020) incorporate auxiliary information about the items (columns). They assume descriptions of each item exists. They use pre-computed word-embeddings to train a neural network to learn user representations. They use a Poisson likelihood with a rate computed as the linear combination of the user and item representations.

Other flexible distribution learning methods can be used to specify expressive priors. For instance, Zhou et al. (2020) extend variational autoencoders (VAEs) and use normalizing flows (Papamakarios et al., 2021) to learn flexible priors on the latent embeddings of users and items. They combine user and item auxiliary information with these latent embeddings to learn user and item representation that are then passed through multi-layered perceptrons to generate the parameter of a Bernoulli distribution.

Empirical Bayes (EB) priors have been explored in the context of variational autoencoders (VAEs) (Kingma & Welling, 2013) under the name aggregated posterior, average encoding distribution (Hoffman & Johnson, 2016) or VampPrior (Tomczak & Welling, 2018). The VampPrior learns an amortized posterior and a prior over the latent variables using a shared neural network. It models a prior on the rows only and was used to address posterior collapse (Tomczak & Welling, 2018) or to learn disentangled representations (Kim & Mnih, 2018). Our method focuses on matrix factorization, and we derive EB priors for both row and column latent variables.

# 3 Empirical Bayes priors for probabilistic matrix factorization

Our goal is to develop empirical Bayes (EB) priors for Bayesian matrix factorization models. We will focus here on Poisson matrix factorization (PMF). In Appendix B, we derive EB priors for Gaussian matrix factorization (GMF).

With matrix factorization, the presence of repeated and identically distributed latent variables for each row and each column provides the opportunity to learn their prior distribution from data. This is a form of empirical Bayes (Robbins, 1992; Efron, 2012) that prescribes a *population prior* (see Section 3.1).

This population prior aims to align the model's marginal distribution of observations with the observed population distribution. In the special case of matrix factorization, there are two distinct populations: the populations of row vectors, and the population of column vectors. With TwinEB, we learn one prior for each population. This is a form of hierarchical modeling without introducing an extra layer of latent variables.

**Notations**. We employ **boldface** symbols to distinguish vectors from scalars. We use the notations:

- $N$ the number of rows (e.g., individuals), $D$ the number of columns (e.g., features).

- $\boldsymbol{X} \in \mathbb{N}^{N \times D}$: the observed data matrix; $X_{i,j}$ is the entry in row $i$ and column $j$.

- $P_{\text{row}}^\star(\boldsymbol{X}_{i,:})$ is the unknown distribution of rows vectors, and $P_{\text{col}}^\star(\boldsymbol{X}_{:,j})$ the one of columns.

- $\boldsymbol{U}_i \in \mathbb{R}^L$ and $\boldsymbol{V}_j \in \mathbb{R}^L$ are model latent variables for row $i$ and column $j$; of dimension $L$.

- $\pi_{\text{row}}$ and $\pi_{\text{col}}$ are the prior distribution of the latent variables.

## 3.1 Background: Population priors for simple hierarchical models

In this paper, we assume to have data $\boldsymbol{X}$ that is drawn from an unkown distribution $P^\star$, also called *population distribution*, that we would like to model with a probabilistic model consisting of latent variables $\boldsymbol{Z}$, a prior over the latent variables $\pi$, and a likelihood function $P(\boldsymbol{X} \mid \boldsymbol{Z})$.

A crucial step in Bayesian statistics is the choice of the prior distribution; if done arbitrarily, it can lead to suboptimal posterior inference (Wang et al., 2021). We choose to follow an empirical Bayes principle that prescribes a *population prior* (Hoffman & Johnson, 2016; Tomczak & Welling, 2018). This prior, by design, aligns the model's marginal distribution of observations with the population distribution $P^\star(\boldsymbol{X})$.

We first focus on a family of latent variable models called simple hierarchical models (Agrawal & Domke, 2021). The joint distribution factorizes as follows:

$$P(\boldsymbol{Z}, \boldsymbol{X}) = \prod_{i=1}^{N} \pi(\boldsymbol{z}_i) P(\boldsymbol{x}_i \mid \boldsymbol{z}_i), \tag{4}$$

where $\pi$ is the prior distribution of $\boldsymbol{z}_i$. To simplify notations, we then focus on the marginal likelihood of a single observation, $\boldsymbol{x}$, and its corresponding local latent variable $\boldsymbol{z}$.

An empirical Bayes criterion is that the marginal distribution of observations under the model, denoted as $P_\pi(\boldsymbol{x})$, should match with their true population distribution $P^\star(\boldsymbol{x})$ (Ignatiadis & Wager, 2022), that is:

$$\begin{aligned} P^\star(\boldsymbol{x}) &= P_\pi(\boldsymbol{x}) \\ &= \int \pi(\boldsymbol{z}) P(\boldsymbol{x} \mid \boldsymbol{z}) \, d\boldsymbol{z}. \end{aligned} \tag{5}$$

Our goal is to set $\pi$ such that Equation (5) holds. The expression for the prior $\pi$ that satisfies this conditions is:

$$\begin{aligned} \pi(\boldsymbol{z}) &\approx \int P_\pi(\boldsymbol{z} \mid \boldsymbol{x}) P^\star(\boldsymbol{x}) d\boldsymbol{x} \\ &= \mathbb{E}_{P^\star(\boldsymbol{x})}[P_\pi(\boldsymbol{z} \mid \boldsymbol{x})], \end{aligned} \tag{6}$$

where $P_\pi(\boldsymbol{z} \mid \boldsymbol{x})$ is the (local) posterior distribution of the latent variable $\boldsymbol{z}$ given the observation $\boldsymbol{x}$ under the model. The definition presents two issues: the unknown true population distribution $P^\star(\boldsymbol{x})$, and the fact that the target prior $\pi$ is on both sides of Equation (6), explicitly on the left and implicitly via the posterior on the right. The research literature has approximated Equation (6) with Monte Carlo estimates of $P^\star(\boldsymbol{x})$ and variational inference of $P_\pi(\boldsymbol{z} \mid \boldsymbol{x})$ (Hoffman & Johnson, 2016; Tomczak & Welling, 2018).

### 3.2 Population priors for probabilistic matrix factorization

Our goal is to develop population priors for Bayesian matrix factorization models. The challenge is that unlike simple hierarchical models, there is no distinction between local and global latent variables, rather latent variables denote row- and column-specific random variables.

#### 3.2.1 Twin population priors

We establish population priors for two latent variables, one for the latent variables of the rows $\pi_{\text{row}}(\boldsymbol{U}_i)$ and one for the latent variables of the columns $\pi_{\text{col}}(\boldsymbol{V}_j)$. These priors will match two different populations, one of the row vectors and one of the column vectors:

$$P^\star_{\text{row}}(\boldsymbol{X}_{i,:}) \coloneqq \text{Population distribution of row vectors}$$
$$P^\star_{\text{col}}(\boldsymbol{X}_{:,j}) \coloneqq \text{Population distribution of column vectors}$$

We recall that the populations distributions are unkown distributions, from which we sample the rows and the columns.

We begin with population priors for row latent variables. As in section 3.1, we specify the prior based on an empirical Bayes principle such that the true marginal distribution of the rows $P^\star_{\text{row}}(\boldsymbol{X}_{i,:})$ is aligned with the distribution of the rows under the model, that is:

$$P^\star_{\text{row}}(\boldsymbol{X}_{i,:}) = \int \pi_{\text{row}}(\boldsymbol{U}_i) \prod_{j=1}^{D} P(X_{i,j} \mid \boldsymbol{U}_i, \boldsymbol{V}_j) d\boldsymbol{U}_i, \tag{7}$$

for a fixed set of column variables $\boldsymbol{V}$. For Equation (7) to hold, a population prior should be used:

$$\pi_{\text{row}}(\boldsymbol{U}_i; \boldsymbol{V}) = \mathbb{E}_{P^\star_{\text{row}}(\boldsymbol{X}_{i,:})}[P_{\pi_{\text{row}}}(\boldsymbol{U}_i \mid \boldsymbol{X}_{i,:}, \boldsymbol{V})], \tag{8}$$

where we explicted the dependence of the EB prior on the column latent variables $\boldsymbol{V}$. Similarly for columns, for a fixed set of row latent variables $\boldsymbol{U}$, the empirical Bayes criterion is:

$$P^\star_{\text{col}}(\boldsymbol{X}_{:,j}) = \int \pi_{\text{col}}(\boldsymbol{V}_i) \prod_{i=1}^{N} P(X_{i,j} \mid \boldsymbol{U}_i, \boldsymbol{V}_j) d\boldsymbol{V}_j. \tag{9}$$

The prior that satisfies this criterion is the column population prior:

$$\pi_{\text{col}}(\boldsymbol{V}_j; \boldsymbol{U}) = \mathbb{E}_{P^\star_{\text{col}}(\boldsymbol{X}_{:,j})}[P_{\pi_{\text{col}}}(\boldsymbol{V}_j \mid \boldsymbol{X}_{:,j}, \boldsymbol{U})]. \tag{10}$$

Since there are two populations in need of prior specification, we call Equations (8 and 10) the *twin* population priors.

We have established the form of population priors in probabilistic matrix factorization. Next, we focus on how to estimate the twin population priors and how to approximate posterior inference under them.

In the remainder of the paper, we focus on the Poisson matrix factorization (PMF). We derive the priors for Gaussian matrix factorization (GMF) in Appendix B.

### 3.3 Twin EB prior for Poisson matrix factorization

In Poisson matrix factorization (PMF), the row and column latent variables $\boldsymbol{U}_i$ and $\boldsymbol{V}_i$ are non-negative $L$-vectors and the likelihood in Equation (3) is Poisson:

$$X_{i,j} \mid \boldsymbol{U}_i, \boldsymbol{V}_j \sim \text{Poisson}\left(\boldsymbol{U}_i^T \boldsymbol{V}_j\right), \tag{11}$$

for $i \in [N], j \in [D]$.

The log-likelihood of the data is:

$$
\begin{aligned}
\log P(\boldsymbol{X} \mid \boldsymbol{U}, \boldsymbol{V}) &= \log \prod_{i=1}^{N} \prod_{j=1}^{D} P(X_{i,j} \mid \boldsymbol{U}_i, \boldsymbol{V}_j) \\
&= \sum_{i,j}^{N,D} \log \text{Poisson}\left(X_{i,j} \mid \boldsymbol{U}_i^T \boldsymbol{V}_j\right).
\end{aligned} \tag{12}
$$

Some methods place Gamma priors on $\boldsymbol{V}_j$ and $\boldsymbol{U}_i$ (Gopalan et al., 2015). Note that this is a Bayesian formulation of non-negative matrix factorization (Cemgil, 2009).

To compute the population prior for the rows, $\pi_{\text{row}}(\boldsymbol{U}_i; \boldsymbol{V}) = \mathbb{E}_{P^{\star}_{\text{row}}(\boldsymbol{X}_{i,:})}[P_{\pi_{\text{row}}}(\boldsymbol{U}_i \mid \boldsymbol{X}_{i,:}, \boldsymbol{V})]$, we face two problems. Namely, we do not know the true population distribution of the rows $P^{\star}_{\text{row}}(\boldsymbol{X}_{i,:})$, and the population prior $\pi_{\text{col}}(\boldsymbol{U}_i)$ appears on both sides of the equality.

To find the population prior, we first notice that a Monte Carlo estimate of Equation (8) writes as:

$$\pi_{\text{row}}(\boldsymbol{U}_i; \boldsymbol{V}) \approx \frac{1}{N} \sum_{i'=1}^{N} P_{\pi_{\text{row}}}(\boldsymbol{U}_i \mid \boldsymbol{X}_{i',:}, \boldsymbol{V}). \tag{13}$$

When the prior satisfies Equation (13), this property is called self-consistency (Laird, 1978).

The structure of Equations (13) suggests to use families of mixtures of parametric distributions to approximate the row and column population priors (Tomczak & Welling, 2018). Mixtures can approximate complex distributions when their number of components increases while having the convenience of remaining parametric (Titterington et al., 1985; Nguyen et al., 2020). We choose to model the priors by dropping their dependence on the other variable and express them as,

$$\pi_{\text{row}}(\boldsymbol{U}_i) := P_{\theta_{\text{row}}}(\boldsymbol{U}_i) := \sum_{k=1}^{K_r} \omega_k P_{\boldsymbol{\mu}_k, \boldsymbol{\sigma}_k}(\boldsymbol{U}_i), \tag{14}$$

$$\pi_{\text{col}}(\boldsymbol{V}_j) := P_{\theta_{\text{col}}}(\boldsymbol{V}_j) := \sum_{k=1}^{K_c} \rho_k P_{\boldsymbol{\nu}_k, \boldsymbol{\eta}_k}(\boldsymbol{V}_j), \tag{15}$$

where $K^r$ and $K^c$ are the number of components in the mixtures, $\theta_{\text{row}} = \{\boldsymbol{\mu}, \boldsymbol{\sigma}, \boldsymbol{\omega}\}$ and $\theta_{\text{col}} = \{\boldsymbol{\nu}, \boldsymbol{\eta}, \boldsymbol{\rho}\}$. The locations $\boldsymbol{\mu} \in \mathbb{R}^{K_r \times L}$ and $\boldsymbol{\nu} \in \mathbb{R}^{K_c \times L}$, the scales $\boldsymbol{\sigma} \in \mathbb{R}^{K_r \times L}$ and $\boldsymbol{\eta} \in \mathbb{R}^{K_c \times L}$, and the mixture weights $\omega \in \Delta^{K_r}$ and $\rho \in \Delta^{K_c}$ are the parameters of the mixtures priors. As $K_r$ and $K_c$ increase, the priors become more and more expressive (see Figure 4). Figure 1 shows a graphical model representation of matrix factorization with EB priors.

In a classical empirical Bayes setup, the idea is to set the priors that maximize the marginal likelihood of the data:

$$\hat{\theta}_{\text{row}}, \hat{\theta}_{\text{col}} = \underset{\theta_{\text{row}}, \theta_{\text{col}}}{\arg\max} \log P(\boldsymbol{X}; \theta_{\text{row}}, \theta_{\text{col}}), \tag{16}$$

where

$$\log P(\boldsymbol{X}; \theta_{\text{row}}, \theta_{\text{col}}) = \log \int P_{\theta_{\text{row}}}(\boldsymbol{U}) P_{\theta_{\text{col}}}(\boldsymbol{V}) P(\boldsymbol{X} \mid \boldsymbol{U}, \boldsymbol{V}) \, d\boldsymbol{U} d\boldsymbol{V}. \tag{17}$$

In Section 3.4, we find $\theta_{\text{row}}, \theta_{\text{col}}$ at the same time as we approximate the model posterior $P(\boldsymbol{U}, \boldsymbol{V} \mid \boldsymbol{X}; \theta_{\text{row}}, \theta_{\text{col}})$.

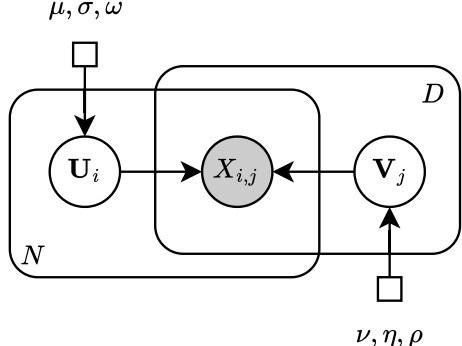

**Figure 1: Twin population priors for Poisson matrix factorization model**. Shaded nodes are observed while other nodes represent latent random variables. The empty squares indicate that we will fit these priors to the data.

### 3.4 Posterior inference in PMF with twin EB priors

Given data $\boldsymbol{X}$, our goal is to calculate the posterior $P(\boldsymbol{U}, \boldsymbol{V} \mid \boldsymbol{X}; \theta_{\text{row}}, \theta_{\text{col}})$, which also depends on our choice of priors $\theta_{\text{row}}, \theta_{\text{col}}$. The challenges are that this posterior is intractable (for any prior) and we simultaneously want to fit the priors to satisfy the EB criterion in Equation (16).

Our strategy will be as follows. We will use variational inference (VI) (Blei et al., 2017) to approximate the posterior, taking gradient steps in the variational objective with respect to the posterior approximation (the variational family). At the same time, however, the variational objective of VI is an approximation (lower bound) of the log-marginal from Equation (16). So we also take gradient steps with respect to the EB priors to maximize it. The result is an algorithm that simultaneously approximates the posterior and learns the EB prior.

**The variational posterior.** Consider a parameterized mean-field variational family,

$$q_{\boldsymbol{\Lambda}}(\boldsymbol{U}, \boldsymbol{V} \mid \boldsymbol{X}) = \prod_{i,l} q_{\boldsymbol{\lambda}_{i,l}^r}(U_{i,l}) \prod_{j,l} q_{\boldsymbol{\lambda}_{j,l}^c}(V_{j,l}), \tag{18}$$

This family has parameters for each row's latent vector and each column's latent vector, $\boldsymbol{\lambda}_i^r$ and $\boldsymbol{\lambda}_i^c$ respectively. We further define $\boldsymbol{\Lambda}^r := [\boldsymbol{\lambda}_{i,l}^r]$, and $\boldsymbol{\Lambda}^c := [\boldsymbol{\lambda}_{j,l}^c]$. The full set of variational parameters is $\boldsymbol{\Lambda} = \{\boldsymbol{\Lambda}^r, \boldsymbol{\Lambda}^c\}$.

From the perspective of posterior inference, our goal is to set $q_{\boldsymbol{\Lambda}}$ to minimize the KL divergence to the exact posterior:

$$\hat{\boldsymbol{\Lambda}} = \arg\min_{\boldsymbol{\Lambda}} \text{KL}(q_{\boldsymbol{\Lambda}}; P(\boldsymbol{U}, \boldsymbol{V} \mid \boldsymbol{X}; \theta_{\text{row}}, \theta_{\text{col}})). \tag{19}$$

In detail, the variational family is a bank of Log-Normals:

$$\boldsymbol{\lambda}_{i,l}^r := (a'_{i,l}, b'_{i,l}), \tag{20}$$

$$\boldsymbol{\lambda}_{j,l}^c := (a_{j,l}, b_{j,l}), \tag{21}$$

$$q_{\boldsymbol{\lambda}_{i,l}^r}(U_{i,l}) := \mathcal{LN}(a'_{i,l}, b'_{i,l}), \tag{22}$$

$$q_{\boldsymbol{\lambda}_{j,l}^c}(V_{j,l}) := \mathcal{LN}(a_{j,l}, b_{j,l}). \tag{23}$$

Each Log-Normal is parameterized by its natural parameters $a$ and $b$:

$$\mathcal{LN}(x; a, b) \propto \exp\left(-\frac{a}{2b}\log(x) - \frac{(\log x)^2}{2b}\right).$$

To minimize the KL divergence in Equation (19), VI optimizes the variational parameters $\mathbf{\Lambda}$ to, equivalently, maximize the evidence lower bound (ELBO) (Blei et al., 2017):

$$
\begin{aligned}
\mathcal{L}(\boldsymbol{X}; \mathbf{\Lambda}, \theta_{\mathrm{row}}, \theta_{\mathrm{col}}) = \mathbb{E}_{q_{\mathbf{\Lambda}}(\boldsymbol{U}, \boldsymbol{V} \mid \boldsymbol{X})}&[\log P(\boldsymbol{X} \mid \boldsymbol{U}, \boldsymbol{V})] \\
&+ \mathbb{E}_{q_{\mathbf{\Lambda}}}[\log P(\boldsymbol{U}; \theta_{\mathrm{row}}, \theta_{\mathrm{col}}) + \log P(\boldsymbol{V}; \theta_{\mathrm{row}}, \theta_{\mathrm{col}})] \\
&- \mathbb{E}_{q_{\mathbf{\Lambda}}}[\log q_{\mathbf{\Lambda}}(\boldsymbol{U}, \boldsymbol{V} \mid \boldsymbol{X})]. \quad (24)
\end{aligned}
$$

Here, we use gradient ascent to maximize $\mathcal{L}(\boldsymbol{X}; \boldsymbol{\theta}, \mathbf{\Lambda})$ with respect to $\mathbf{\Lambda}$ (Ranganath et al., 2014). We further use stochastic reparameterization gradients to take such steps (Kingma & Welling, 2013; Rezende et al., 2014).

**Maximum marginal likelihood.** At the same time, we would like to set the prior parameters to maximize the marginal likelihood of the data (Equation (16)). The variational objective in Equation (24) conveniently also provides a lower-bound on the marginal likelihood (Blei et al., 2017):

$$
\log P(X; \theta_{\mathrm{row}}, \theta_{\mathrm{col}}) \geq \mathcal{L}(\boldsymbol{X}; \boldsymbol{\theta}, \mathbf{\Lambda}). \tag{25}
$$

So, we will also follow stochastic gradients of the ELBO with respect to the prior parameters $\theta_{\mathrm{row}}, \theta_{\mathrm{col}}$ to maximize $\mathcal{L}(\boldsymbol{X}; \boldsymbol{\theta}, \mathbf{\Lambda})$ with respect to $\theta_{\mathrm{row}}, \theta_{\mathrm{col}}$. This strategy has been used in the context of linear regression (Mukherjee et al., 2023).

**Twin EB.** Putting these two pieces together, our algorithm is a stochastic gradient ascent of the ELBO with respect to two sets of parameters. In optimizing with respect to $\mathbf{\Lambda}$, we minimize the KL divergence between $q_{\mathbf{\Lambda}}$ and the posterior; in optimizing with respect to $\theta_{\mathrm{row}}, \theta_{\mathrm{col}}$, we maximize the (approximate) marginal likelihood of the data.

We use the Adam algorithm for stochastic optimization (Kingma & Ba, 2014) with a batch size of 128, and using ten particles to obtain unbiased noisy estimates of the gradient of the ELBO via the reparameterization trick (the particles are samples from $q_{\mathbf{\Lambda}}$ to estimate the expectation $\mathbb{E}_{q_{\mathbf{\Lambda}}}$ of the ELBO with Monte-Carlo).

The details of the algorithm are in Algorithm 1. Our implementation is available at `https://github.com/blei-lab/TwinEB`.

---

**Algorithm 1** Variational inference for Poisson matrix factorization with twin EB priors

---

**Input:** Data $\boldsymbol{X}$, number of particles $S$, learning rate $\zeta$, number of iterations $T$, number of components $K_r, K_c$, number of latent dimensions $L$.
**Output:** Variational posterior parameters $\mathbf{\Lambda}^*$, prior parameters $\theta_{\mathrm{row}}{}^*, \theta_{\mathrm{col}}{}^*$.
**Initialize:** $\mathbf{\Lambda}^{(0)}, \theta_{\mathrm{row}}{}^{(0)}, \theta_{\mathrm{col}}{}^{(0)}$.
**for** $t = 1$ to $T$ **do**
  **for** $i = 1$ to $N$, $l = 1$ to $L$ **do**
    **for** $s = 1$ to $S$ **do**
      Sample $\epsilon_{i,l}^{(s)} \sim \mathcal{N}(0, 1)$.
      Compute $U_{i,l}^{(s)} = \exp(a'{}_{i,l}^{(t-1)} + b'{}_{i,l}^{(t-1)} \epsilon_{i,l}^{(s)})$.
    **end for**
  **end for**
  **for** $j = 1$ to $D$, $l = 1$ to $L$ **do**
    **for** $s = 1$ to $S$ **do**
      Sample $\phi_{j,l}^{(s)} \sim \mathcal{N}(0, 1)$.
      Compute $V_{j,l}^{(s)} = \exp(a_{j,l}^{(t-1)} + b_{j,l}^{(t-1)} \phi_{j,l}^{(s)})$.
    **end for**
  **end for**
  Estimate $\mathcal{L}(\boldsymbol{X}; \theta_{\mathrm{row}}{}^{t-1}, \theta_{\mathrm{col}}{}^{(t-1)}, \mathbf{\Lambda}^{(t-1)})$ using Monte-Carlo in Equation (24) with samples $\boldsymbol{U}^{(s)}, \boldsymbol{V}^{(s)}$ in place of $\mathbb{E}_{q_{\mathbf{\Lambda}}}$.
  $\mathbf{\Lambda}^{(t)}, \theta_{\mathrm{row}}{}^{(t)}, \theta_{\mathrm{col}}{}^{(t)} \leftarrow \mathrm{Adam}(\nabla_{(\theta_{\mathrm{row}}, \theta_{\mathrm{col}}, \mathbf{\Lambda})} \mathcal{L}, \zeta)$.
**end for**
**return** $\mathbf{\Lambda}^{(T)}, \theta_{\mathrm{row}}{}^{(T)}, \theta_{\mathrm{col}}{}^{(T)}$.

---

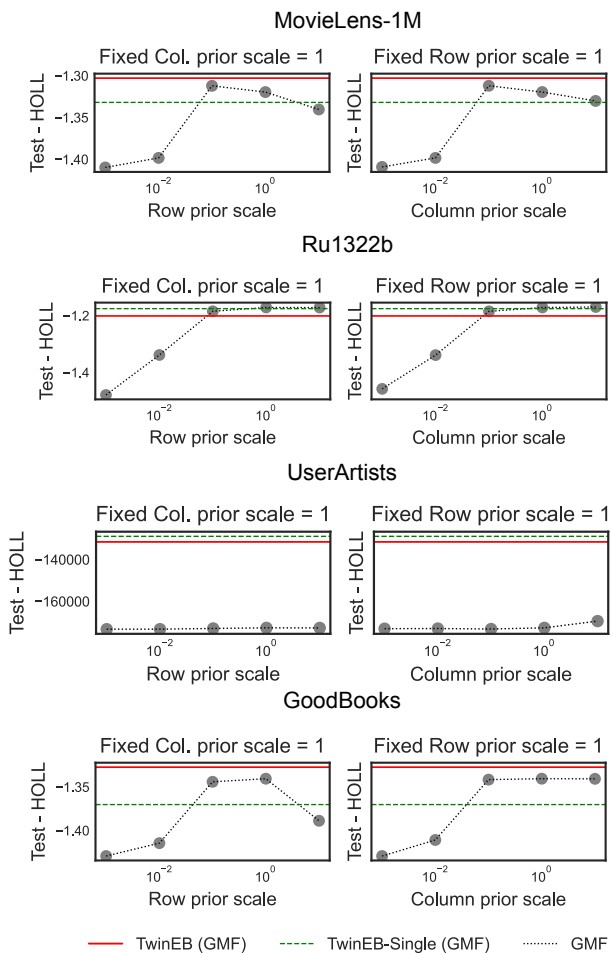

**Figure 2: Twin population priors induce robustness to prior selection** The held-out likelihood is sensitive to the choice of the prior hyper-parameters. GMF endowed with population priors on both row and column latent variables, TwinEB (GMF), achieves comparable or better results than other methods. Each sub-panel displays held-out log-likelihood from adjusting the column prior variance with a fixed row prior, while the right sub-panel does the opposite, varying the row prior variance with a constant column prior. We demonstrate four datasets, from the top to bottom, MovieLens 1M, Ru1322b-scRNAseq, UserArtists, and GoodBooks. In all datasets, we set $L = 15$. Similar results hold for other values of $L$ (see Appendix D.3).

## 4 Experiments

We study Algorithm 1 in several real-world matrix factorization settings: book ratings, movie ratings, artist preferences, and single-cell RNA sequence gene counts. For all datasets, we studied both Gaussian and Poisson matrix factorization. In all datasets, we set $L = 15$. Similar results hold for other values of $L$ (see Appendix D.3). We found that TwinEB performs as well or better than manually searching for a parameterized prior, and performs as well or better than setting a simple parameterized prior by empirical Bayes.

### 4.1 Datasets

In this section we first review four real-world datasets, ranging from user-preferences to genomics, and then explore the impact of twin population priors on the performance of Gaussian and Poisson matrix factorization. In our experiments, we fix the number of row and column mixtures to $K_r = 70$ and $K_c = 100$ respectively.

**MovieLens 1M**. This dataset comprises 1 million ratings from 6,000 users (rows) on 4,000 movies (columns) (Harper & Konstan, 2015). The ratings are on a scale of 1 to 5. Sparsity of this dataset, defined as the number of nonzero elements divided by the total number of entries, is 0.04. We can use matrix factorization to capture different aspects of user preferences and movie characteristics. Specifically, $U_{i,l}$ may signify user $i$'s affinity for aspect $l$ (e.g., genre), while $V_{j,l}$ may represent the degree to which movie $j$ exhibits aspect $l$.

**Ru1322b**. We analyze single cell gene expression data from a patient with small cell lung cancer (Chan et al., 2021). This dataset comprises 4,000 highly variable genes (columns) across 5,308 cells (rows). For the GMF family, we applied a two-step transformation. First, we performed a log transformation on the counts after adding a pseudo-count of one. Then, we standardized the non-zero elements. Sparsity of this dataset is 0.16. Each entry of the matrix denotes the number of transcripts of gene $j$ in cell $i$. We explain the gene expression matrix via $L$ gene-modules, with $U_{i,l}$ as the activity of module $l$ in cell $i$, and $V_{j,l}$ as gene $j$'s contribution to module $l$. Matrix factorization can be used as exploratory data analysis (finding gene modules associated with malignancy in cancer) or as a component in a more complex analysis (causal inference (Wang & Blei, 2019)). See Appendix C for details on preprocessing of the sequencing data.

**UserArtists** We use the data introduced in Cantador et al. (2011), comprising 92,834 user-listened artist relations, across 1,892 users and 17,632 artists, with a maximum value of $352,698$. Similar to MovieLens 1M, we use matrix factorization to explain different aspects of user preferences and artist groupings. Sparsity of this dataset is 0.003. We note that this dataset was analyzed in da Silva et al. (2023).

**GoodBooks** Contains 6 million ratings across 51,288 users and 10,000 books[1]. Ratings range from 1 to 5. Sparsity of this dataset is 0.01.

## 4.2 Evaluation Metric and Baselines

We evaluate model performance using the likelihood of unseen interactions, that is, test holdout-likelihood (HOLL). To build intuition, we will use the example of user-item interactions where rows indicate users and columns items, and the entries of the input matrix their observed interactions. One approach to testing performance is to hold-out a fraction of the entries of the input matrix, and to train the model on the held-in portion, and test performance on the held-out entries. In this approach, all users are observed during training. To test generalizability to cases when new users are observed, we can further hold-out a portion of users (rows) during training, and then test the performance on these hold-out users. Specifically, we will test the performance on masked entries of these hold-out users. This procedure measures *strong generalization* (Steck, 2019).

At test time, we need to learn user-specific latents for the held-out rows. For this, we run the model in a special training step where the column latents are held fixed. Since these held-out users are nevertheless coming from the same pool as our training users, we use only 30% of the observed entries for this special training step. This may result in a more accurate measurement of out-of-distribution generalizability by breaking spurious correlations between items and users due to biases present in the current dataset. These biases may include the time-period of data collection, platforms used that may not exist in a new dataset. Finally, we report the model performance on 30% of the unseen entries of the held-out users. See the supplement for the derivation of Equation (45) and more details on the experiments.

**Baseline**. We evaluate the performance of the PMF (GMF) model with TwinEB against multiple baselines, namely (i) TwinEB-Single, (ii) PMF (GMF). TwinEB-Single is a simple form of TwinEB where all latent dimensions have an identical prior, which we learn. PMF is a prior of the same family as the TwinEB, but with fixed hyperparameters. We compare TwinEB against a large choice of fixed parameters, akin to hyperparameter selection.

---

[1]Accessed at https://github.com/zygmuntz/goodbooks-10k

### 4.3 Experimental Procedure and Results: Gaussian Matrix Factorization

We preprocess the data as follows. We standardize each column by subtracting the mean from non-zero entries and dividing the result by their standard deviation. We study two scenarios: (i) maintaining a fixed prior on row-wise variables while varying the prior on column-wise variables, and (ii) holding the prior on column-wise variables constant and adjusting the prior on row-wise variables. We set the variational family as well as the mixture components for the population priors to be Gaussian. We set the fixed prior to $\mathcal{N}(0,1)$ and vary the variance of the non-fixed one over $\{0.001, 0.01, 0.1, 1.0, 10\}$. For example, in Figure 2, the left column of plots corresponds to fixing the prior on the column-wise variables to $P(\boldsymbol{V}_j) = \mathcal{N}(0,1)$ and varying the prior on the row-wise variables as $P(\boldsymbol{U}_i) = \mathcal{N}(0, \sigma^2)$ with $\sigma^2 \in \{0.001, 0.01, 0.1, 1, 10\}$.

We compared the GMF model to TwinEB (GMF) and TwinEB-Single (GMF). We treat zero entries as missing values.

Figure 2 displays the outcomes for four real-world datasets. In GoodBooks and MovieLens-1M datasets, TwinEB achieves the highest test HOLL. For the UserArtists dataset, TwinEB does better than the fixed prior. Similar results were obtained by varying $L$ to other values and are reported in Supplementary Figures 5, 6 in Appendix D.

We also compare to the method of Wang & Stephens (2021), we used the corresponding package flashr, that currently supports GMF. We designed an imputation experiment where we held-out 10% of entries of a standardized matrix, and compare reconstruction accuracy on non-zero entries. We applied flashr to the above four real-world datasets. Except in Ru1322b, the optimization objective in flashr encounters NaN values, which halt execution and terminate without results. For the Ru1322b dataset it achieves a mean absolute error of 0.64 vs 0.51 for the TwinEB. For a comparison on simulated data, please see Supplementary Table S1. We note that it is not straightforward to compare our method to that of Zhong et al. (2022); the software implementation does not immediately support missing data and imputation.

### 4.4 Experimental Procedure and Results: Poisson Matrix Factorization

For the MovieLens 1M and GoodBooks datasets, we binarize the matrix, setting entries to 'one' if a user has rated a movie and to 'zero' otherwise. For the Ru1322b dataset, we normalize the rows such that the sum of all rows are equal; we then round each value to the nearest integer. This is to account for the effect of library size (Heumos et al., 2023). We treat zeros as missing data. In all models, we set the variational family to be Log-Normal, and the prior (or the mixture components for the population priors) to be Gamma. Similar to GMF, we vary the row or column prior parameters, while keeping the other one fixed. We parameterize the Gamma prior by its mean and variance $\text{Gamma}(\mu, \sigma^2)$, where $\mu = \alpha/\beta$ and $\sigma^2 = \alpha/\beta^2$. In each scenario, we set the fixed prior to $\text{Gamma}(1, 10)$. For the varying prior, we set its mean to $\mu = 1$ and change its variance along $\{0.01, 0.1, 0.25, 1.0, 10.0, 100.0\}$. Figure 3 shows the results.

In datasets except for GoodBooks, TwinEB outperforms TwinEBSingle. In the UserArtists dataset, TwinEB outperforms other methods. TwinEB, on average, takes about 1.5X the runtime of PMF (ranging form 1.1X to 2.1X). We point out that finding a good prior using grid-search incurs the sum of the cost of evaluations of the individual points in the grid, over 6.5X the running time of the PMF with TwinEB prior. We note that using a grid to find fixed priors (or alternatively, learning the scalar prior) does not guarantee reaching a best test HOLL. Indeed in the UserArtists dataset, TwinEB yields the best test HOLL.

We also compared TwinEB to the method of da Silva et al. (2023); given the data, it estimates values for the shape and rate parameters of the Gamma prior for both the row and column r.v.s. Table 1 displays the estimated hyperparameters and the resulting test HOLL.

PMF equipped with fixed priors values calculated from this methods yields Test HOLL that are on par or slightly worse than the TwinEB method. This method yielded negative hyperparameter estimates for the MovieLens-1M and GoodBooks datasets. These parameters are effectively not usable, since the parameters of a Gamma must be positive. Hence, we emphasize that unlike ours, this methods cannot be used on all datasets. We explore additional baselines in Appendix D.2.

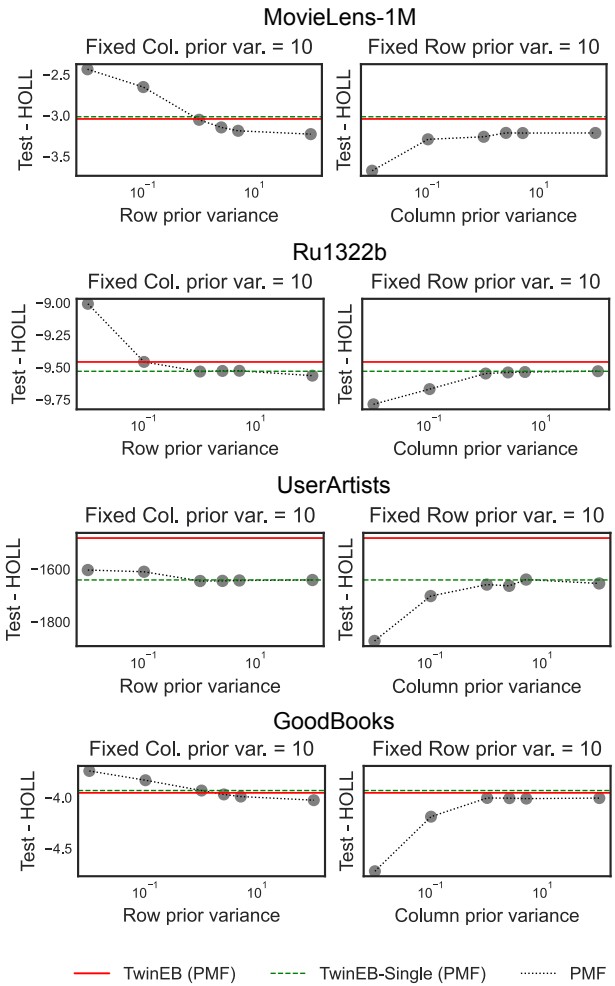

**Figure 3: Twin population priors induce robustness to prior selection**. As in Figure 2, but for PMF.

**Table 1:** Test Heldout Loglikelihood from running PMF with prior hyperparameters sets using the method of da Silva et al. (2023). For MovieLens-1M and GoodBooks datasets, da Silva et al. (2023) yielded inadmissible hyperparameters.

|  | test HOLL | |
|---|---|---|
| Dataset | da Silva et al. (2023) | TwinEB |
| Ru1322b | -9.53 | **-9.46** |
| UserArtists | -1,583 | **-1,481** |
| MovieLens-1M | NA | **-3.04** |
| GoodBooks | NA | **-3.95** |

## 4.5 Simulation: Complexity of the Prior

Here, we examine the performance of the population priors as the number of mixture components are varied. To this end, we simulate a $1,000$ by $1,500$ dataset, with $L = 64$ dimensional row and column-wise r.v.s. We sample the row- and column-wise r.v.s from a mixture of 15 and 20 Gamma distributions respectively, the

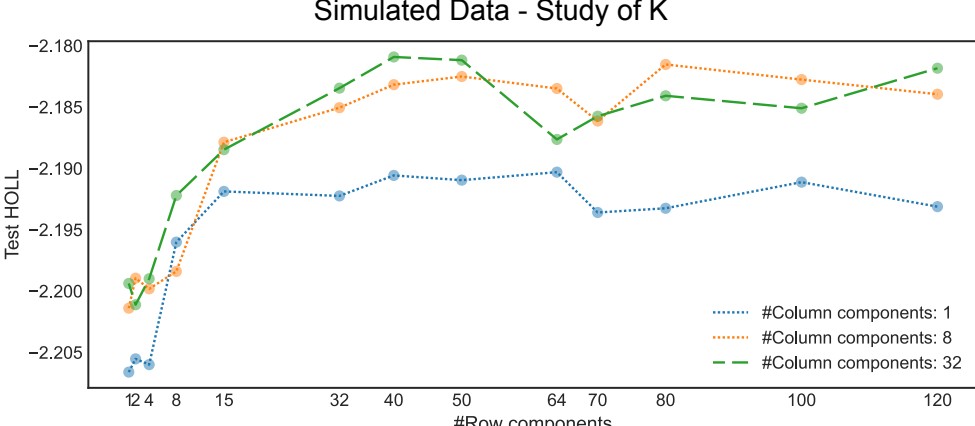

**Figure 4: Increasing the number of mixtures improves the model performance**. Each line shows the average test-HOLL over 10 seeds for a fixed value for $K_c$, the number of column mixtures, and varying values of $K_r$, the number of row mixtures. Here, we set $L = 64$ to its true simulated value. The results are similar for $L = 32$ (see supplement).

rate and shape parameters of which are sampled from a $\text{Gamma}(1, 1)$. The mixture weights are sampled from a $\text{Dirichlet}(e_0, \ldots, e_0)$ where the concentration parameter is $e_0 = 10$.

**Performance Increase with Number of Mixture Components.** The more mixture components, the more flexible the EB prior is. If learning a EB prior is beneficial, then we expect the performance to increase with the number of mixture components. We run PMF with population priors with varying number of row and column mixtures, setting the dimension of the latent variable to its true value, and, to avoid model misspecification, set the mixture prior family to Gamma. We then report the average of the test HOLL across ten different seeds. Figure 4, the log-likelihood of test held-out data increases with the number of mixture components in the row prior.

## 5 Discussion

We introduced the twin population priors for probabilistic matrix factorization. We derived a method to estimate the corresponding posterior using Monte Carlo and variational inference. On real-world data, this method finds a prior as good as the best parametric prior chosen retrospectively.

Our method uses the widely applicable black box variational inference (BBVI); the availability and ease of use of automatic differentiation, our implementation brings flexible priors in the context of matrix factorization to a wider audience. For example, our proposed method works on multiple data types, both Gaussian and Poisson likelihoods, and it is easy to extend it to other reparameterizable likelihoods. In contrast, the methods of Wang et al. (2024) and da Silva et al. (2023) are tailored for only Gaussian and only count data only.

With TwinEB, the latent dimensions $L$ of the row-and column-wise variables is a hyperparameter. In practice, we found that setting $L$ at about 15 was generally good enough (used across all our experiments) and ablation studies showed that the results were stable with the choice of $L$ (GFM: Supplementary Figure 5 , PMF: Supplementary Figure 6). However, an attractive property of the method of da Silva et al is that it can automatically set this value in closed form. Methods based on prior predictive statistics suggested by da Silva et al. (2023) could optimize the selection of $L$.

A limitation of our model compared to others, is that it does not leverage auxiliary information. Auxiliary information is information in addition to the observed interaction matrix. For instance, for movie ratings, the additional information could be user-specific attributes including age, race, and occupation, while item-

specific attributes could include description of the items. Methods like Wang et al. (2024) and Van Linh et al. (2020) propose ways to integrate auxiliary information to improve the inference in matrix factorization.

We use a linear function to model the dependence of the observed interactions on the per-row per-column latent variables. This has the advantage of being interpretable and prevent overfitting (e.g., in the analysis of single cell gene expression data (Svensson et al., 2020)). However, in the presence of non-linear interactions, this model would be misspecified. In that case, the link function could be changed, e.g., Zhou et al. (2020) use a multi-layered perceptron (MLP) to model the dependence of the observed interactions on the per-row per-column latent variables.

Other flexible distribution learning methods can be used to specify expressive priors, such as normalizing flows used in Zhou et al. (2020), and one of our baseline methods, TwinNF (Appendix D.2). The model begins with a simple base distribution and iteratively transforms it into a complex distribution that can capture more intricate patterns in the data.

One area of further work is to extend this algorithm to tensor factorization (Kolda & Bader, 2009; Schein et al., 2015). While in matrix factorization, each entry of the observed matrix is explained via two latent variables, tensor factorization models will involve more. One detail to address is how to formally define the population distribution associated with each latent variable.

Investigating theoretical properties of the twin population prior a la da Silva et al. (2023) would offer insights into the behavior of the priors before data is observed that can be a possible future research direction.

**Broader Impact Statement**

As a data-driven approach, our method is subject to potential pitfalls inherent in such methods, including a tendency to inherit and amplify biases present in the data. If employed without proper supervision, this can lead to serious consequences including biased and/or discriminatory outcomes in real-world applications. These biases implicitly discriminate based on protected attributes (e.g., race, gender). One line of research assumes specific protected features are known, and enforces fairness by decoupling these features from learnt latent variables (Togashi & Abe, 2022). If fairness criteria can be formulated in terms of a regularization term, it can be reinterpreted as learning fairness-aware priors and incorporated in our framework (Zhu et al., 2018).

## 6 Acknowledgements

This research was funded in part through the NIH/NCI Cancer Center Support Grant P30 CA008748 (list all MSK authors); NIH grant 5K99CA277562-02 (S.S.), the Eric and Wendy Schmidt Center at the Broad Institute of MIT and Harvard (A.N.), the Africk Family Fund (A.N.), the National Science Foundation (NSF) grants IIS-2127869 and DMS-2311108 (D.B.), the Office of Naval Research grant N000142412243 (D.B). and the Simons Foundation (D.B.). SPS was partially supported by the Halvorsen Center for Computational Oncology and the MacMillan Center for the Non-Coding Cancer Genome.

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

# A  Introduction

This is the supplement for the manuscript titled "Population priors for matrix factorization". The figures in this document supplement Figures 2, 3, and 4 in the main text. We provide derivations for twin population priors for Gaussian matrix factorization, as well as computing the likelihood of the held-out data in section B. We then give more details about our experimental setup, including the parameters used in training (e.g., batch-size) in section C. Finally, we provide results for additional experiments in section D: for additional values of the latent dimension $L$ for the datasets studied in the main text.

Note that this document is accompanied by an archive file `code.zip`, that contains the source code, instructions to install and run the code, and scripts to recreate the experiments and plot the figures in the manuscript. All scripts have been de-identified.

# B  Derivations

In this section, we give more details and derivations for the quantities defined in the main text. Specifically, in section B.1 we derive the twin population priors for the Gaussian matrix factorization (GMF). In section B.2, we derive the variational approximation to the twin population priors in GMF. Finally, in section B.3, we derive the expression for held-out likelihood in matrix factorization.

## B.1  Twin population priors for Gaussian matrix factorization

In this section we derive population priors for Gaussian matrix factorization (GMF). In the classical GMF, the likelihood and priors on the latent variables are Gaussian (Mnih & Salakhutdinov, 2007). The generative model is as follows:

$$
\begin{aligned}
W_{j,l} &\sim \mathcal{N}(0, \sigma_W^2), \quad j = 1 \ldots D, l = 1 \ldots L, \\
Z_{i,l} &\sim \mathcal{N}(0, \sigma_Z^2), \quad i = 1 \ldots N, l = 1 \ldots L, \\
X_{i,j} \mid \boldsymbol{U}_i, \boldsymbol{V}_j &\sim \mathcal{N}(\sum_{l=1}^{L} Z_{i,l} W_{jl}, \sigma^2), \quad i = 1 \ldots N, j = 1 \ldots D,
\end{aligned}
\tag{26}
$$

where $\boldsymbol{U} = [Z_{i,l}] \in \mathbb{R}^{N \times L}$ and $[W_{j,l}] \in \mathbb{R}^{D \times L}$, with $\boldsymbol{U}_i$ and $\boldsymbol{V}_j$ and are $L$-vectors representing the row and column wise latent variables, and $\sigma$, $\sigma_W$ and $\sigma_Z$ are constant.

For TwinEB, our goal is to learn the prior for the row and column latent variables, so, as in the main text, we construct mixture priors (Equations 14 and 15), where $P_{\boldsymbol{\mu}_k, \boldsymbol{\sigma}_k}$ and $P_{\boldsymbol{\nu}_k, \boldsymbol{\eta}_k}$ are Gaussian parameterized by their mean and scale parameters. That is,

$$
P_{\theta^r}(\boldsymbol{U}_i) = \sum_{k=1}^{K_r} \omega_k \mathcal{N}(\boldsymbol{U}_i; \boldsymbol{\mu}_k, \boldsymbol{\sigma}_k),
\tag{27}
$$

$$
P_{\theta^c}(\boldsymbol{V}_j) = \sum_{k=1}^{K_c} \rho_k \mathcal{N}(\boldsymbol{V}_j; \boldsymbol{\nu}_k, \boldsymbol{\eta}_k),
\tag{28}
$$

$$
\tag{29}
$$

where $K^r$ and $K^c$ are the number of components in the mixtures, $\theta^r = \{\boldsymbol{\mu}, \boldsymbol{\sigma}, \boldsymbol{\omega}\}$ and $\theta^c = \{\boldsymbol{\nu}, \boldsymbol{\eta}, \boldsymbol{\rho}\}$. The locations $\boldsymbol{\mu} \in \mathbb{R}^{K_r \times L}$ and $\boldsymbol{\nu} \in \mathbb{R}^{K_c \times L}$, the scales $\boldsymbol{\sigma} \in \mathbb{R}^{K_r \times L}$ and $\boldsymbol{\eta} \in \mathbb{R}^{K_c \times L}$, and the mixture weights $\boldsymbol{\omega} \in \Delta^{K_r}$ and $\boldsymbol{\rho} \in \Delta^{K_c}$ are the parameters of the mixtures priors.

## B.2  Posterior inference in GMF with twin population priors

Our goal is to calculate the posterior $P(\boldsymbol{U}, \boldsymbol{V} \mid \boldsymbol{X}.\theta^r, \theta^c)$, given data $\boldsymbol{X}$, which also depends on our choice of priors $\theta^r, \theta^c$. Our strategy is the same as in the main text, namely, to simultaneously optimize for the

parameters of the variational distribution and the TwinEB prior. We substitute the variational family in Equations (19) and (19) with Gaussian, as follows:

$$\boldsymbol{\lambda}_{i,l}^{r} \coloneqq (a'_{i,l}, b'_{i,l}), \tag{30}$$

$$\boldsymbol{\lambda}_{j,l}^{c} \coloneqq (a_{j,l}, b_{j,l}), \tag{31}$$

$$q_{\boldsymbol{\lambda}_{i,l}^{r}}(U_{i,l}) \coloneqq \mathcal{N}(a'_{i,l}, b'_{i,l}), \tag{32}$$

$$q_{\boldsymbol{\lambda}_{j,l}^{c}}(V_{j,l}) \coloneqq \mathcal{N}(a_{j,l}, b_{j,l}). \tag{33}$$

Each Gaussian is parameterized by its natural parameters $a$ and $b$:

$$\mathcal{N}(x; a, b) \propto \exp\left(ax + bx^2\right)$$

The log-likelihood of the data is:

$$
\begin{aligned}
\log P(\boldsymbol{X} \mid \boldsymbol{Z}, \boldsymbol{W}) &= \log \prod_{i=1}^{N} \prod_{j=1}^{D} P_\sigma(X_{i,j} \mid \boldsymbol{Z}_i, \boldsymbol{W}_j) \\
&= \sum_{i=1}^{N} \sum_{j=1}^{D} \log \mathcal{N}\left(X_{i,j}; \sum_{l=1}^{L} Z_{i,l} W_{j,l}, \sigma^2\right) \\
&= -\sum_{i=1}^{N} \sum_{j=1}^{D} \left[\log \sigma\sqrt{2\pi} + \frac{1}{2}\left(\frac{X_{i,j} - \sum_{l=1}^{L} Z_{i,l} W_{j,l}}{\sigma}\right)^2\right].
\end{aligned}
\tag{34}
$$

### B.3 Held-out likelihood for matrix factorization

We use the log likelihood of held-out data as the score for each model. Let $X^{\text{out}} = \{X_{i,j}\}$ denote $N^{\text{out}}$ entries of the held-out rows that were masked. We compute the likelihood of the masked entries via the posterior predictive distribution:

$$\log P(X^{\text{out}} \mid X^{\text{in}}) = \log \prod_{i=1}^{N^{\text{out}}} P(X_i^{\text{out}} \mid X^{\text{in}}) \tag{35}$$

$$= \sum_{i=1}^{N^{\text{out}}} \log P(X_i^{\text{out}} \mid X^{\text{in}}). \tag{36}$$

For Bayesian matrix factorization we expand the summand in Equation (36) as follows:

$$\log P(X_i^{\text{out}} \mid X^{\text{in}}) \approx \log \iint P(X_i^{\text{out}} \mid \boldsymbol{U}, \boldsymbol{V}) P(\boldsymbol{U}, \boldsymbol{V} \mid X^{\text{in}}) d\boldsymbol{U} d\boldsymbol{V} \tag{37}$$

$$\approx \log \iint P(X_i^{\text{out}} \mid \boldsymbol{U}, \boldsymbol{V}) q_{\boldsymbol{\Lambda}}(\boldsymbol{U}, \boldsymbol{V} \mid X^{\text{in}}) d\boldsymbol{U} d\boldsymbol{V} \tag{38}$$

$$= \log \mathbb{E}_{\boldsymbol{U}, \boldsymbol{V} \sim q_{\boldsymbol{\Lambda}}(.)}[P(X_i^{\text{out}} \mid \boldsymbol{U}, \boldsymbol{V})] \tag{39}$$

$$\approx \log \frac{1}{M} \sum_{m}^{M} P(X_i^{\text{out}} \mid \boldsymbol{U}^{(M)}, \boldsymbol{V}^{(M)}), \tag{40}$$

where $M$ is the number of Monte Carlo samples from $q_{\boldsymbol{\Lambda}}(\boldsymbol{U}, \boldsymbol{V} \mid X^{\text{in}})$. In Equation (38) we approximate the true posterior $P(\boldsymbol{U}, \boldsymbol{V} \mid X^{\text{in}})$ with its variational counterpart $q_{\boldsymbol{\Lambda}}$. The log-likelihood score for the entire held-out data is then:

$$\sum_{i}^{N^{\text{out}}} \log P(X_i^{\text{out}} \mid X^{\text{in}}) \approx \sum_{i}^{N^{\text{out}}} \log \frac{1}{M} \sum_{m}^{M} P(X_i^{\text{out}} \mid \boldsymbol{U}^{(M)}, \boldsymbol{V}^{(M)}). \tag{41}$$

## C   Experimental details

In this section we give more details on our experimental studies in the main manuscript. In section C.1 we describe preprocessing for the gene expression in the Ru1322-scRNAseq dataset. In section C.2, we specify the parameters used during training. In section C.4 we give a brief description of the artifacts that accompany this supplementary material.

### C.1   Preprocessing of the Ru1322-scRNAseq gene expression dataset

We use CellRanger version `6.0.1` to process the FASTQ and generate the unique molecular identifier (UMI) count matrices. We use `scanpy` to preprocess the data, and the `seuratv3` algorithm to select highly variable genes Wolf et al. (2018).

### C.2   Training details

We set a batch size of 128 in all our experiments. We ran Poisson and Gaussian matrix factorization experiments for a maximum of $20,000$ iterations. By this step, all runs had converged.

We initialized the learning rate for the row and column variables, `rlr` and `clr` separately. We fix the initial learning rate $\texttt{rlr} \in \{0.01\}$ and $\texttt{clr} \in \{0.01\}$. In the experiments in the main text, we use 10 Monte Carlo samples to approximate the ELBO, while in the supplemental experiments, we use a single particle.

For the PMF experiments in the supplement, we subsample zeros as is standard (Gopalan et al., 2015). We uniformly randomly subsample the same number of zeros as non-zero values to estimate the likelihood. Concretely, let $\mathcal{L}(X^{\text{out}})$ denote the log-likelihood of the masked entries of the held-out rows, $X^{\text{out}} = \{X_{i,j}\}$. Then $\mathcal{L}(X^{\text{out}})$ can be decomposed as the sum of non-zero and zero $X_{i,j}$:

$$\mathcal{L}(X^{\text{out}}) = \mathcal{L}(X^{\text{out}}_{x_{i,j} \neq 0}) + \mathcal{L}(X^{\text{out}}_{x_{i,j}=0}), \tag{42}$$

where $X^{\text{out}}_{x_{i,j} \neq 0}$ and $X^{\text{out}}_{x_{i,j}=0}$ have cardinalities $\text{N}^{\text{out}}_{\text{non-zero}}$ and $\text{N}^{\text{out}}_{\text{zero}}$ respectively.

We approximate Equation (42) by subsampling $\text{N}_{\text{sub}} = \min(\text{N}^{\text{out}}_{\text{zero}}, \text{N}^{\text{out}}_{\text{non-zero}})$ of the zeros. Let $\boldsymbol{X}^{\text{zero}}_{\text{N}_{\text{sub}}}$ denote a multiset of zeros of cardinality $\text{N}_{\text{sub}}$, then:

$$\hat{\mathcal{L}}(X^{\text{out}}) \approx \mathcal{L}(X^{\text{out}}_{x_{i,j} \neq 0}) + \frac{\text{N}^{\text{out}}_{\text{zero}}}{\text{N}_{\text{sub}}} \boldsymbol{X}^{\text{zero}}_{\text{N}_{\text{sub}}}. \tag{43}$$

We ran our experiment on a machine equipped with an `NVIDIA A100` GPU with 80GB memory. We implemented all methods in pytorch (Paszke et al., 2019).

### C.3   Evaluation

We evaluate each model based on the likelihood of held-out data.

For a given row-wise data split into held-in and held-out rows, let $X^{\text{out}} = \{X^{\text{out}}_i\}$ denote the set of $N^{\text{out}}$ masked entries of the held-out rows. We estimate the held-out log-likelihood as follows:

$$\frac{1}{N^{\text{out}}} \sum_{i=1}^{N^{\text{out}}} \log P(X^{\text{out}}_i \mid X^{\text{in}}) \approx \tag{44}$$

$$\frac{1}{N^{\text{out}}} \sum_{i=1}^{N^{\text{out}}} \log \frac{1}{M} \sum_{m=1}^{M} P(X^{\text{out}}_i \mid \boldsymbol{U}^{(m)}, \boldsymbol{V}^{(m)}), \tag{45}$$

where $M$ is the number of Monte Carlo samples from $q_{\boldsymbol{\Lambda}}(\boldsymbol{U}, \boldsymbol{V})$.

We randomly assign 20% of the rows as the test set, and the rest as training data. We then mask 20% of the entries at random, and train the model on this train set using ten random restarts. We use the 20% masked

entries as a *validation set*. At test time, we put aside 30% of the entries of the test rows at random - these entries constitute the *test set* - then we train the model on 40% of the rest of the entries. This procedure measures *strong generalization* (Steck, 2019).

To compute the held-out likelihood, after training the model on the held-in data, we fix its column parameters and then learn row parameters of the held-out rows. We then report the likelihood of the masked entries in the test set. For each model, we report the test HOLL of the random restart that achieved the best validation HOLL. In our experiments, we set $M = 500$.

### C.4 Implementation

Please refer to the `code` directory for the source code and instructions on how to run the model. The file `README.md` contains instructions for installing the software, and running it. Under the `data` directory, we included preprocessed data for the MovieLens-100K dataset, a smaller version of the MovieLens-1M studied in the main text. In the `notebooks` directory, we put notebooks used for preprocessing the data, and plotting the figures in the manuscript. Finally, in the `pipelines` directory, we put `nextflow` scripts that recreate experiments that we have run for the manuscript.

## D Additional experiments

In this section we present additional experimental results. We show results for additional values of the latent dimension $L$. Concretely, section D.3 studies the effects of the twin population priors on Poisson and Gaussian matrix factorization, on three real world datasets, while section D.4 examines the sensitivity of the twin population priors to the choice of its hyper-parameters. We find that these results corroborate those that were presented in the main manuscript, that is, matrix factorization with traditional priors is sensitive to the choice of the hyper-parameters of the prior, and twin population priors is a robust way to set the prior in this family of models.

### D.1 Simulated studies

We compare our method to that of Wang & Stephens (2021) on a synthetic dataset, namely that used in Figure 4 in the main manuscript. Table S1 shows the mean absolute error of non-zero entries, averaged over 10 different repeats. The results are comparable, where TwinEB does slightly better.

**Table S1:** Comparison of methods based on Mean Absolute Error.

| Method | Mean Absolute Error |
|---|---|
| Wang & Stephens (2021) | 0.37 |
| TwinEB (GMF) | **0.35** |

### D.2 Additional baselines

Here, we introduce two additional baseline methods. One is based on normalizing flows (Papamakarios et al., 2021), and the other is based on learning the hyper-parameters of Gamma priors for our PMF experiments (close to Hierarchical Poisson Factorization of Gopalan et al. (2015)). We call both Twin, since we are simultaneously learning the hyperparameters of the priors on the row and column latents.

**TwinHP** In TwinHP, we infer the hyper-parameters of the two Gamma priors. This represents a more exhaustive search in the hyper-parameter space of a model with uni-modal priors. Table S2 shows the results (a positive value indicates that TwinEB performs better). We note that in the Ru1322b dataset this method has the worst performance (as measured by Test Hold-Out likelihood), suggesting that the uni-modal prior is not well specified for that dataset. On the other hand, it achieves the best HOLL in the MovieLens-1M dataset, suggesting that a uni-modal prior is more appropriate in this dataset.

**Table S2:** Comparison of TwinEB to TwinHP in Test Heldout Loglikelihood (Delta-Test HOLL) across various real-world datasets. Delta-Test HOLL = Test $\text{HOLL}_{\text{TwinEB}}$ − Test $\text{HOLL}_{\text{TwinHP}}$.

| Dataset | Delta-Test HOLL |
|---|---|
| Ru1322b | **0.54** |
| UserArtists | **211.74** |
| MovieLens-1M | -0.61 |
| GoodBooks | **0.01** |

**TwinNF** We implement a Normalizing Flow (NF) (Papamakarios et al., 2021) prior by stacking 10 Planar flows (similar to the method of Zhou et al. (2020)) to define a flexible prior distribution. Each Planar flow applies an affine transformation to the latent variables $\mathbf{z}$ through the mapping:

$$\mathbf{f}_z = \mathbf{z} + \mathbf{u} \cdot \tanh(\mathbf{w}^T \mathbf{z} + b),$$

where $\mathbf{u}$, $\mathbf{w}$ are learnable parameters and $b$ is a bias term. The determinant of the Jacobian is computed as:

$$\det(\nabla \mathbf{f}_z) = \left| 1 + \mathbf{u}^T \left( (1 - \tanh^2(\mathbf{w}^T \mathbf{z} + b)) \cdot \mathbf{w} \right) \right|.$$

We do not claim that this implementation cannot be improved. Nevertheless, we find that TwinEB compares well with this NF prior, out-performing it in 3 out of the 4 real-world datasets. We show the result in the following table, where the second column shows the difference between Test-HOLL of TwinEB and TwinNF (a positive value indicates that TwinEB performs better). We have used the same training procedure as in the main text, with 10 seeds.

**Table S3:** As in Table S2, but for TwinNF. Delta-Test HOLL = Test $\text{HOLL}_{\text{TwinEB}}$ − Test $\text{HOLL}_{\text{TwinNF}}$.

| Dataset | Delta-Test HOLL |
|---|---|
| Ru1322b | -0.12 |
| UserArtists | **350.46** |
| MovieLens-1M | **0.18** |
| GoodBooks | **0.40** |

### D.3 Additional values of latent dimensions

In the main text, we study the effects of twin population priors on the Poisson and Gaussian matrix factorization over four real datasets, with the dimension of the latent variables set to $L = 15$. Here, we present results for additional values of $L$.

### D.4 Performance Increases, then Plateaus with the Number of Mixture Components,

We show the performance of TwinEB on simulated data as we vary the number of row and column mixture components . Here, we add results for additional values of $L$, and also study the MovieLens 100K dataset. The result in Figures 7 suggests that as the number of mixture components increases, the performance of TwinEB improves. The gain is apparent when the number of row components $K_r$ increases. This is expected, as our evaluation procedure, measures the generalizability for held-out rows, but not held-out columns.

In Figure 8 we show the performance on real-world datasets as we vary the number of row and column mixtures. Again we find that, overall, as the performance first increases, until plateauing.

## E Extended literature review

Here we review, in detail, the methods of Wang & Stephens (2021) and da Silva et al. (2023). These methods are closest to ours in methodology.

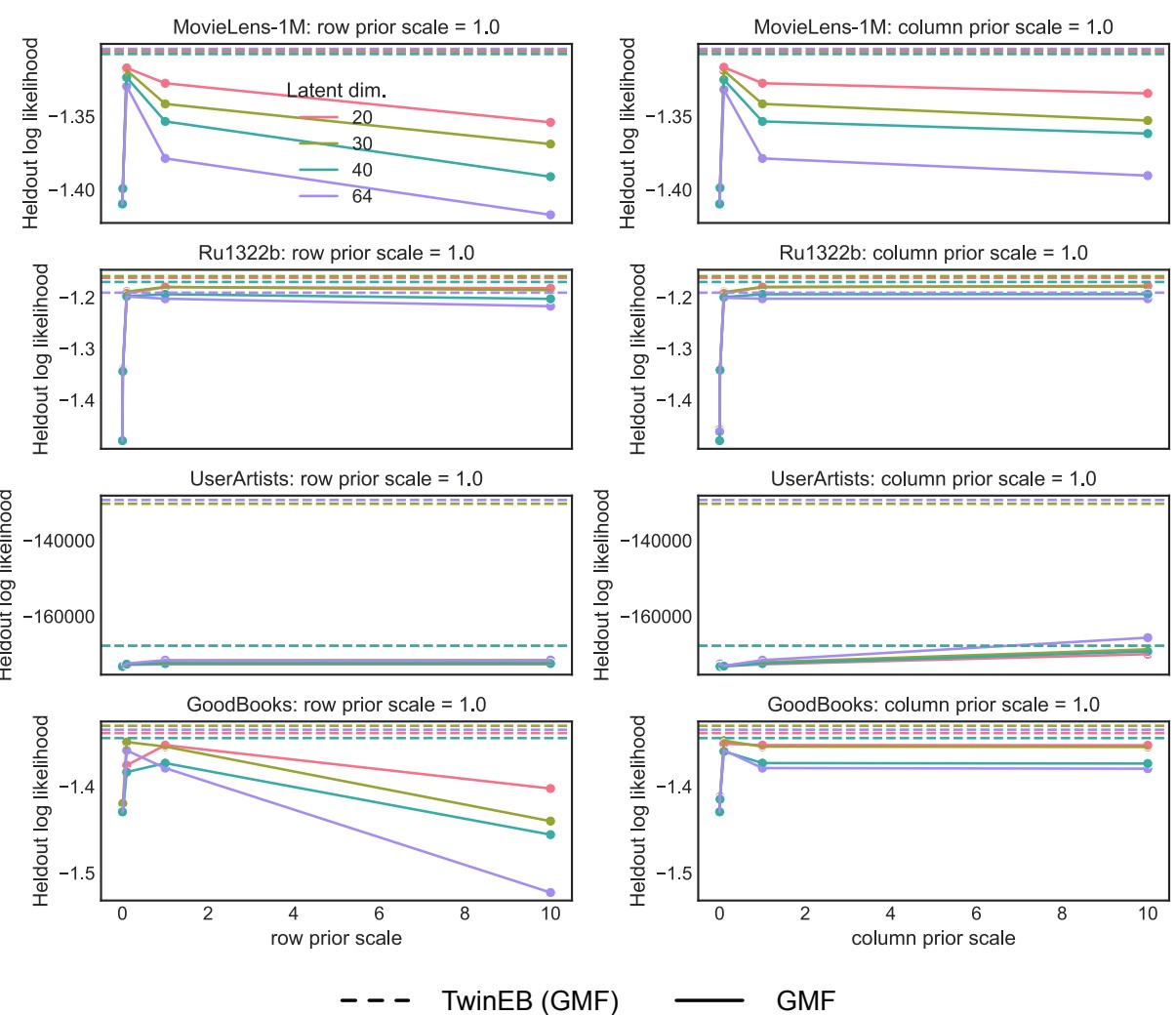

**Figure 5:** As in Figure 2, but with additional values of latent dimension $L$.

### E.1 Method of Wang & Stephens (2021)

Given the following model for observations $X = [X_{i,j}] \in \mathbb{R}^{N \times D}$ we can find the row and column factors by solving a empirical Bayes normal means (EBNM) problem:

$$
\begin{aligned}
X_{i,j} &= \sum_{l=1}^{L} u_{i,l} v_{j,l} + E_{i,j} \\
u_{i,l} &\sim g_{u_l} \in \mathcal{G}_u \\
v_{j,l} &\sim g_{v_l} \in \mathcal{G}_v \\
E_{i,j} &\sim \mathcal{N}(0, 1/\tau_{i,j}).
\end{aligned}
\tag{46}
$$

We set $\tau_{i,j} = 1$ for all $i, j$ to simplify notation.

To solve for $\mathbf{u}$ and $\mathbf{v}$, empirical Bayes (EB) prescribes that (1) we learn the $g_u$ and $g_v$ by maximizing the marginal log likelihood of the data, and (2) then return inference of $u$ and $v$ amounts to learning the induced posterior by $\hat{g}_u$ and $\hat{g}_v$.

We use mean-field variational EM, to solve this problem.

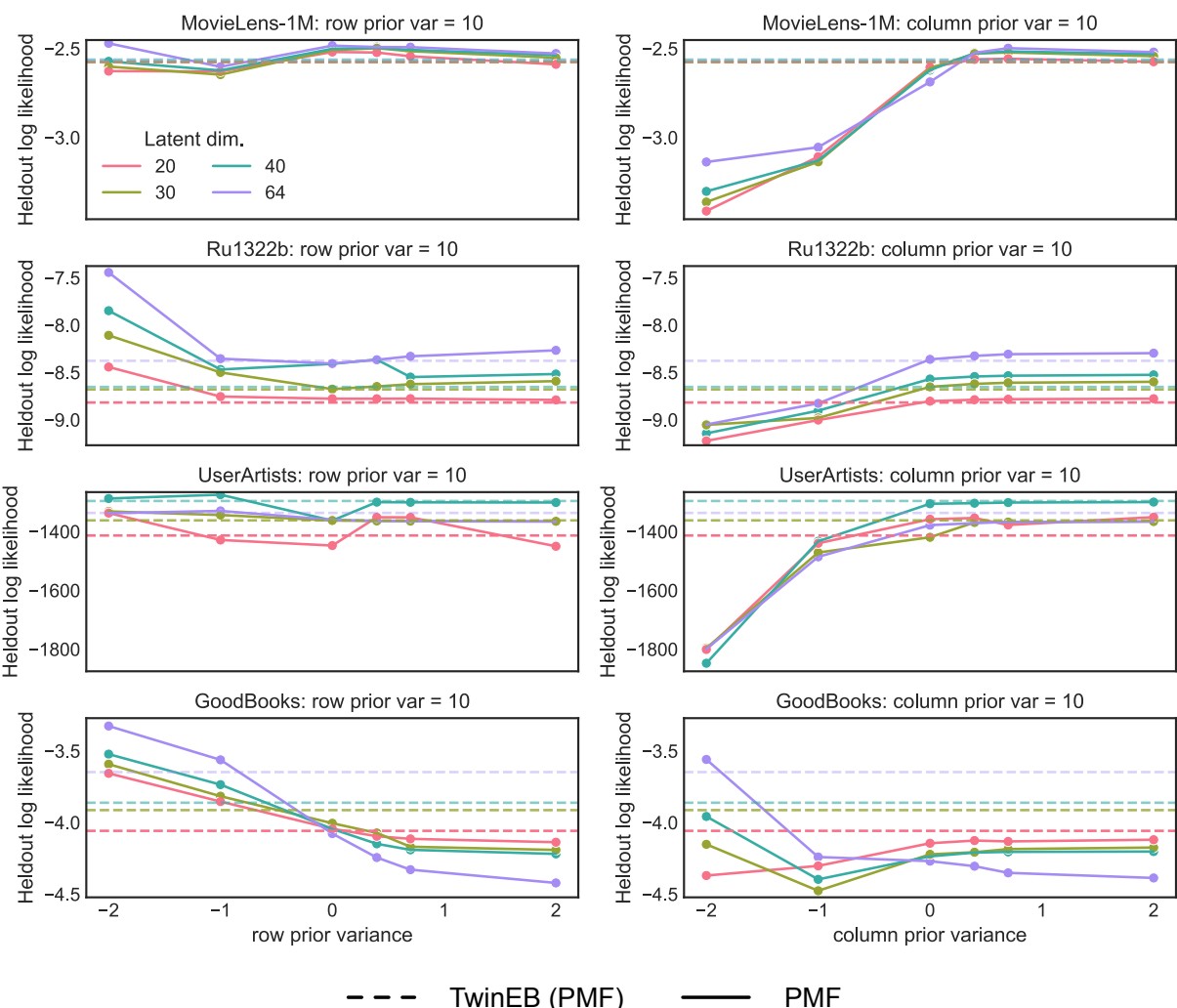

**Figure 6:** As in Figure 3, but with additional values of latent dimension $L$.

Define the ELBO, $F(q_{\mathbf{u}}, g_{\mathbf{u}}, q_{\mathbf{v}}, g_{\mathbf{v}})$ our objective to be maximized as follows:

$$F(q_{\mathbf{u}}, g_{\mathbf{u}}, q_{\mathbf{v}}, g_{\mathbf{v}}) = \int q(\mathbf{u}, \mathbf{v}) \log \frac{p(X, \mathbf{u}, \mathbf{v} \mid g_{\mathbf{u}}, g_{\mathbf{v}})}{q(\mathbf{u}, \mathbf{v})} d\mathbf{u} d\mathbf{v} \tag{47}$$

Assume $L$, the dimension of the row and column factors is 1. Initialize $g_u$, $g_v$, $u$, and $v$, set $t = 0$ and until convergence do:

$$t \leftarrow t + 1 \tag{48}$$

$$q_u^{(t)}, g_u^{(t)} \leftarrow \underset{q_u, g_u}{\arg\max} F(q_u, g_u, q_v^{(t-1)}, g_v^{(t-1)}) \tag{49}$$

$$q_v^{(t)}, g_v^{(t)} \leftarrow \underset{q_v, g_v}{\arg\max} F(q_u^{(t)}, g_u^{(t)}, q_v, g_v) \tag{50}$$

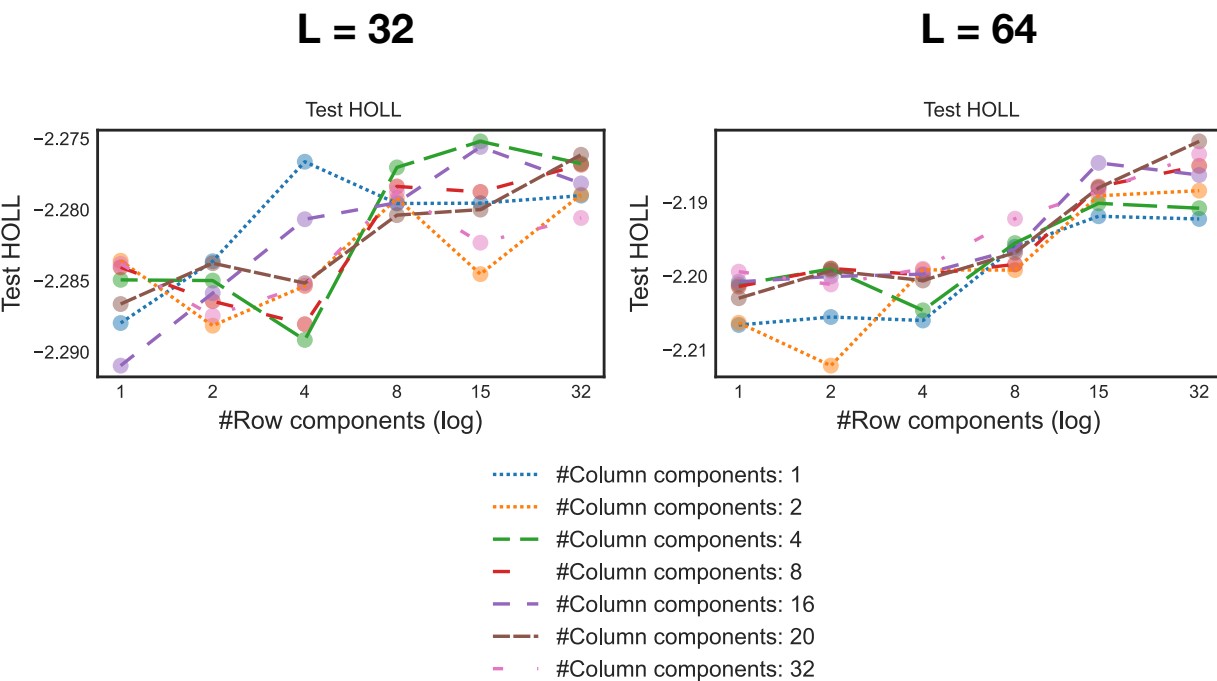

**Figure 7:** As in Figure 4, but with additional column components as well as latent dimension $L = 32$.

### E.2 Empirical Bayes normal means

For known $\mathbf{s} = [s_i]$, observations $\mathbf{x} = [x_j]$ are assumed to be distributed as follows:

$$
\begin{aligned}
\theta &\sim g, g \in \mathcal{G} \\
x_i \mid \theta_i &\overset{\text{iid}}{\sim} \mathcal{N}(x_i; \theta_i, s_i).
\end{aligned}
\tag{51}
$$

Empirical Bayes procedure prescribes a two step process for inference over $\theta = [\theta_j]$:

1. Find a prior $g(.)$ that maximizes the marginal likelihood of the data $\mathcal{L}(\mathbf{x})$:

$$
\hat{g} = \arg\max_{g'} \mathcal{L}(\boldsymbol{x}) = \arg\max_{g'} \prod_{i=1}^{N} \int g'(\theta_i) p_{s_i}(x_i \mid \theta_i) d\theta_i.
\tag{52}
$$

2. Compute the induced posteriors by $\hat{g}$:

$$
p_{\mathbf{s}}(\theta \mid \mathbf{x}, \hat{g}) \approx \prod_{i=1}^{N} \hat{g}(\theta_i) p_{s_i}(x_i \mid \theta_i).
\tag{53}
$$

Solving this problem can be seen as a mapping from the observations $\mathbf{x}$ and $\mathbf{s}$ to the estimated prior $\hat{g}$ and induced posteriors $p_{\mathbf{s}}(\theta \mid \mathbf{x})$, that is:

$$
(\hat{g}(\theta), p_{\mathbf{s}}(\theta \mid \mathbf{x})) \leftarrow \text{EBNM}(\mathbf{x}, \mathbf{s})
\tag{54}
$$

It turns out that in the variational EM algorithm, only the first two moments of the induced posterior are necessary, so we define them:

$$
\begin{aligned}
\bar{\theta}_j &= \mathbb{E}_{\mathbf{s}}[\theta_j \mid \mathbf{x}, \hat{g}] \\
\bar{\theta^2}_j &= \mathbb{E}_{\mathbf{s}}[\theta_j^2 \mid \mathbf{x}, \hat{g}].
\end{aligned}
\tag{55}
$$

### E.3 Main proposition

$$\arg\max_{g_u, q_u} F(q_u, g_u, q_v^{(t-1)}, g_v^{(t-1)}) = \text{EBNM}(\hat{\mathbf{u}}(Y, \bar{\mathbf{v}}, \bar{\mathbf{v}^2}), \hat{\mathbf{s}}(\bar{\mathbf{v}^2})), \tag{56}$$

where

$$\hat{\mathbf{u}}(Y, \boldsymbol{h}, \boldsymbol{w}) = \frac{\sum_j^D Y_{ij} h_j}{\sum_j^D w_j} \tag{57}$$

$$\hat{\mathbf{s}}(\boldsymbol{w}) = \left(\sum_j^D \boldsymbol{w}_j\right)^{-0.5}. \tag{58}$$

Intuitively, the $\hat{u}_i$ is normally distributed centered at $u_i$, since for a fixed $v$ we have:

$$\begin{aligned} X_{i,:} &= u_i \boldsymbol{v} + E_{i,:} \Rightarrow \\ \hat{u}_i &= X_{i,:} \boldsymbol{v}^T (\boldsymbol{v}\boldsymbol{v}^T)^{-1} \\ \hat{u}_i &= \frac{\sum_j^D X_{ij} v_j}{\sum_j^D v_j^2}, \end{aligned} \tag{59}$$

where we have computed $i$ independent regressions of the $D$-vector $Y_{i,:}$ on the $D$-vector $\mathbf{v}$. Given the normally distributed noise $E$, $\hat{u}_i \mid u_i \sim \mathcal{N}(\hat{u}_i; u, 1)$.

### E.4 Proof sketch

We show that maximizing the prior in EBNM is maximizing some $F^{NM}$ objective. We then show that by completing the square, we are solving the EBNM problem.

For the EBNM model of Equation (51), define the ELBO:

$$F^{NM}(q_\theta, g) = \int q_\theta(\theta) \log \frac{p(\mathbf{x}, \theta, \mid g)}{q_\theta(\theta)} d\theta. \tag{60}$$

Note that since

$$F^{NM}(q_\theta, g) = \log p_{\mathbf{s}}(\mathbf{x} \mid g) - \text{KL}\left((\parallel q)_\theta, p_{\mathbf{s}}(\theta \mid \mathbf{x}, g), \right. \tag{61}$$

then

$$\max_{q_\theta} F^{NM}(q_\theta, g) = \log p(\mathbf{x} \mid g), \tag{62}$$

$$\arg\max_g \max_{q_\theta} F^{NM}(q_\theta, g) = \arg\max_g \log p(\mathbf{x} \mid g). \tag{63}$$

Now we show that the complete log likelihood of the data in the EBNM model has a squared form:

$$\log p_{\mathbf{s}}(\mathbf{x}, \theta \mid g) = -\frac{1}{2} \sum_{j=1}^D s_j^{-2} (x_j - \theta_j)^2 + \log g(\theta) + \text{const.} \tag{64}$$

Therefore

$$F^{NM}(q_\theta, g) = \mathbb{E}_{q_\theta}\left[-\frac{1}{2} \sum_{j=1}^D (A_j \theta_j^2 - 2B_j \theta_j)\right] + \mathbb{E}_{q_\theta} \log \frac{g(\theta)}{q_\theta(\theta)} + \text{const.} \tag{65}$$

Now to find the updates in Equation (50), we write down $F(.)$ (the ELBO), then complete the square for $\mathbf{u}$ ($\mathbf{v}$) as needed to find the form of $x_j = \frac{B_j}{A_j}$ for the input to the EBNM problem. That is where functions $\hat{\mathbf{u}}(.)$ and $\hat{\mathbf{s}}(.)$ come from.

In their experiments, they set $\mathcal{G}_u = \mathcal{G}_v = \mathcal{G}$, and try two settings for the family of the prior, namely, $\mathcal{G} \in \{SN, PN\}$, the scale mixture of normals, and a mixture of point mass at zero and a standard normal distribution. $\mathcal{G} = SN = \sum_b^B \pi_b \mathcal{N}(0, \sigma_b)$ and $\mathcal{G} = PN = \pi_0 \delta_0 + (1 - \pi_0)\mathcal{N}(0, \sigma_b)$. For $SN$, $\sum_b^B \pi_b = 1$ and $\sigma_b$ are B values lying on a user specified grid.

This paper is somewhat sparse in details. For instance, the *scale mixture of normals* is not explicitly defined anywhere in the paper. More details are found in the first author's PhD thesis Wang (2017). The framework is implemented in the flashr R package.

Extensions to Binary, Binomial and Poisson data are explored in the first author's thesis Wang (2017) as follows.

### E.5 Extension to binary observations

The trick is to reparameterize the mean as log-odds, and then use a lower-bound trick for ELBO Jaakkola & Jordan (2000).

Assuming that $X_{i,j} \in \{-1, +1\}$:

$$
\begin{aligned}
X_{ij} &= 2 \times \text{Bern}(p_{ij}) - 1 \\
p_{ij} &= p(X_{i,j} = 1 \mid z_{ij}) \\
\log \frac{p_{ij}}{1 - p_{ij}} &= z_{ij} \\
z_{ij} &= u_i v_j \\
u_i &\sim g_u(.) \\
v_j &\sim g_v(.).
\end{aligned}
\tag{66}
$$

We write the likelihood as:

$$
P(X_{ij} \mid \mathbf{u}, \mathbf{v}) = P(X_{ij} \mid z_{ij}) = h(Y_{ij}, z_{ij}) = \frac{1}{1 + \exp(-Y_{ij} z_{ij})}
\tag{67}
$$

where $h(z)$ has a (tight) lower bound with parameter $\xi_z$:

$$
\begin{aligned}
h(z) &\geq h(\xi_z) \exp\left(\frac{z - \xi_z}{2} - \tau(\xi_z)(z^2 - \xi_z^2)\right) \\
\tau(\xi_z) &= \frac{1}{2\xi_z}\left(h(\xi_x) - \frac{1}{2}\right) \\
&= \frac{1}{4\xi_z}\tanh(\xi_z/2).
\end{aligned}
\tag{68}
$$

Then replace $P(X \mid \mathbf{u}, \mathbf{v})$ in $F$ with its lower bound $H(\mathbf{u}, \mathbf{v}, X, \xi)$ and call this function $Q(.)$ where $H(.)$ is defined as:

$$
H(\mathbf{u}, \mathbf{v}, X, \xi) = \exp\left\{ \sum_{ij}\left[\frac{X_{ij} u_i v_j - \xi_{ij}}{2} + \log(h(\xi_{ij})) - \tau(\xi_{ij})(u_i^2 v_j^2 - \xi_{ij}^2)\right] \right\},
\tag{69}
$$

where $\xi = [\xi_{ij}] \in \mathbb{R}^{N \times D}$ are variational parameters. This new objective $Q(.)$ has the following form:

$$
Q(\mathbf{u}, \mathbf{v}, X, \xi) = \mathbb{E}_q \log H(\mathbf{u}, \mathbf{v}, X, \xi) + \mathbb{E}_q \log p(\mathbf{u} \mid g_{\mathbf{u}}) p(\mathbf{v} \mid g_{\mathbf{v}}) - \mathbb{E}_q \log q_{\mathbf{u}}(\mathbf{u}) q_{\mathbf{v}}(\mathbf{v}).
\tag{70}
$$

We now maximize $Q(.)$ with the following updates for the EBNM problem:

$$x_j = \frac{0.5 \sum_j (X_{ij} \mathbb{E}_v v_i)}{2 \sum_j (\tau(\xi_{ij}) \mathbb{E}_v v_j^2)},$$

$$s_j^2 = \frac{1}{2 \sum_j (\tau(\xi_{ij}) \mathbb{E}_v v_j^2)}. \tag{71}$$

### E.6 Extension to Binomial observation

For Binomial data, they use a similar generative model to the Bernoulli case, Equation (66), but change the likelihood function to Binomial:

$$X_{ij} = 2 \times \text{Bin}(n_{ij}, p_{ij}). \tag{72}$$

They use an augmentation strategy with the Polya-Gamma random variable. It turns out this augmentation gives a closed form for the expected values of the augmented r.v.s. and the updates are identical to that of the Binary case.

### E.7 Extension to Poisson observation

This is appendix B.2 in the author's PhD thesis. The details are not given.

The generative model is as follows:

$$X_{ij} \mid n_{ij}, p_{ij} = \text{Bin}(n_{ij}, p_{ij})$$

$$\log \frac{p_{ij}}{1 - p_{ij}} = z_{ij} \tag{73}$$

$$n_{ij} \sim \text{Poisson}(\lambda_{ij}).$$

## F   Prior matching da Silva et al. (2023)

**Summary**. Given some target *virtual* statistics (say empirical moments of the observations), (da Silva et al., 2023) picks the hyper-parameters of the prior ($\lambda$) such that the difference between the virtual statistics of the prior predictive distribution and the given virtual statistics are minimized.

Generic Bayesian matrix factorization takes the following form:

$$\text{Row parameters } \theta_{ik} \sim F(\mu_\theta, \sigma_\theta^2),$$

$$\text{Row parameters } \beta_{jk} \sim F(\mu_\beta, \sigma_\beta^2),$$

$$\text{Likelihood } Y_{ij} \sim F_Y \left( \sum_{k=1}^{K} \theta_{ik} \beta_{jk} \right) \tag{74}$$

$$\mathbb{E}[Y_{ij}] = \sum_{k=1}^{K} \theta_{ik} \beta_{jk}.$$

Poisson matrix factorization with Gamma priors takes the following form:

$$\theta_{ik} \stackrel{\text{iid}}{\sim} F(\mu_\theta, \sigma_\theta^2) = \text{Gamma}(a, b),$$

$$\beta_{jk} \stackrel{\text{iid}}{\sim} F(\mu_\beta, \sigma_\beta^2) = \text{Gamma}(c, d),$$

$$Y_{ij} \stackrel{\text{iid}}{\sim} \text{Poisson} \left( \sum_{k=1}^{K} \theta_{ik} \beta_{jk} \right), \tag{75}$$

where $\mu_\theta = \frac{a}{b}$, $\sigma_\theta^2 = \frac{a}{b^2}$, $\mu_\beta = \frac{c}{d}$, $\sigma_\beta^2 = \frac{c}{d^2}$.

A hierarchical Bayesian model $p(Y, Z; \lambda)$, has the following predictive distribution (PPD):

$$p(Y; \lambda) = \int p(Y \mid Z; \lambda) p(Z; \lambda) dZ. \tag{76}$$

**Idea**. Use Equation (76) to integrate out the parameters and search for hyper-parameters that match the data distribution well. Specifically, find a point estimate of $\lambda$ to be used in posterior inference later.

Let $p^\dagger(Y)$ denote the true population distribution of the observations.

$$p^\dagger(Y) = p(Y; \lambda) \tag{77}$$

$$= \int p(Y \mid Z; \lambda) p(Z; \lambda) dZ. \tag{78}$$

Note: in exponential family models with conjugate priors, an analytical form of PPD is available and the marginal likelihood of the data can be directly optimized to find a solution for $\lambda$.

**Algorithm**. Instead of using the true observation distribution $p^\dagger(Y)$, use the observations virtual statistics of $Y$, $\hat{T}_\lambda$.

Find $\lambda$ such that $\hat{T}_\lambda$ match target statistic $T^*$.

In the paper, the authors use moments (mean and variance) of the observed data $Y$. First show closed form solution for PMF. Then provide a more general case based on gradient descent.

### F.1 Analytical solution for Poisson matrix factorization

Given the PMF model in Equation (75), define:

$$\text{Mean} := \mathbb{E}[Y_{ij}; \lambda] \tag{79}$$
$$\text{Variance} := \mathbb{V}[Y_{ij}; \lambda] \tag{80}$$
$$\text{Covariance} := \rho[Y_{ij}, Y_{tl}; \lambda] \tag{81}$$
$$\text{hyper-parameters } \lambda = \{K, \mu_\theta, \sigma_\theta^2, \mu_\beta, \sigma_\beta^2\}. \tag{82}$$

**Proposition 1** *For any entry of the virtual data matrix* $Y = \{Y_{ij}\} \in \mathbb{R}^{N \times M}$*, the mean and variance is given by*:

$$\mathbb{E}[Y_{ij}; \lambda] = K \mu_\theta \mu_\beta \tag{83}$$
$$\mathbb{V}[Y_{ij}; \lambda] = K \left[ \mu_\theta \mu_\beta + (\mu_\beta \sigma_\theta)^2 + (\mu_\theta \sigma \beta)^2 + (\sigma_\theta \sigma_\beta)^2 \right]. \tag{84}$$

**Proposition 2** *For any pair of entries* $Y_{ij}$ *and* $Y_{tl}$ $Y$*, their correlation is given by*:

$$\rho_\lambda[Y_{ij}, Y_{tl}] = \begin{cases} 0 & \text{if } i \neq t \text{ and } j \neq l \\ 1 & \text{if } i = t \text{ and } j = l \\ \rho_1 & \text{if } i = t \text{ and } j \neq l \\ \rho_2 & \text{if } i \neq t \text{ and } j = l \end{cases} \tag{85}$$

*where* $\rho_1 = \frac{K(\mu_\beta \sigma_\theta)^2}{\mathbb{V}_\lambda[Y_{ij}]}$ *and* $\rho_2 = \frac{K(\mu_\theta \sigma_\beta)^2}{\mathbb{V}_\lambda[Y_{ij}]}$.

This yields the following for $K$:

$$K = \frac{\tau \mathbb{V}_\lambda[Y_{ij}] - \mathbb{E}_\lambda[Y_{ij}]}{\rho_1 \rho_2} \left( \frac{\mathbb{E}_\lambda[Y_{ij}]}{\mathbb{V}_\lambda[Y_{ij}]} \right)^2, \tag{86}$$

for $\tau = 1 - (\rho_1 + \rho_2)$.

For the Gamma hyper params, we'll have:

$$
\begin{aligned}
a &= \frac{\rho_2 \mathbb{V}_\lambda[Y_{ij}]}{\tau \mathbb{V}_\lambda[Y_{ij}] - \mathbb{E}_\lambda[Y_{ij}]} \\
c &= \frac{\rho_1 \mathbb{V}_\lambda[Y_{ij}]}{\tau \mathbb{V}_\lambda[Y_{ij}] - \mathbb{E}_\lambda[Y_{ij}]} \\
bd &= \frac{\mathbb{E}_\lambda[Y_{ij}]}{\mathbb{V}_\lambda[Y_{ij}]} \sqrt{\frac{ac}{\rho_1 \rho_2}}.
\end{aligned}
\tag{87}
$$

Given an observed data matrix $Y$, the methods can be used as follows:

1. Compute empirical mean $\hat{\mathbb{V}}_\lambda[Y_{ij}]$ and variance $\hat{\mathbb{E}}_\lambda[Y_{ij}]$ and covariances $\hat{\rho}_\lambda[Y_{ij}, Y_{tl}]$ from $Y$.

2. Replace the virtual statistics in Equations (86 and 87) with their empirical ones to compute the Gamma hyperparameters $a, b, c,$ and $, d$.

Computing $\rho_\lambda[Y_{ij}, Y_{tl}]$ from the single observed matrix $Y$ is not obvious, so we use the following estimator (Algorithm 1 in da Silva et al. (2023)):

1. $A \in \mathbb{R}^{S \times 2}$ and $B \in \mathbb{R}^{S \times 2}$.

2. For $s \in \{1, \dots, S\}$:

3.     Randomly pick a row $i$, then randomly select two distinct indices in that row $j, k$:

4.     $A_{s,1} \leftarrow Y_{i,j}, A_{s,2} \leftarrow Y_{i,k}$,

5.     Randomly pick a column $j$, then randomly select two distinct indices in that column $i, k$:

6.     $B_{s,1} \leftarrow Y_{j,i}, B_{s,2} \leftarrow Y_{j,k}$,

7. $\rho_1 \leftarrow \mathrm{cor}(A_{:,1}, A_{:,2})$ and $\rho_2 \leftarrow \mathrm{cor}(B_{:,1}, B_{:,2})$.

8. Return $\rho_1$ and $\rho_2$

### F.2 Gradient based optimization

When closed form solutions of the moments of the PPD are not available, the hyper-parameter optimization can be done as follows:

$$
\min_\lambda d(T^*, \hat{T}_\lambda),
\tag{88}
$$

for a given (expert determined) $T^*$ and $\hat{T}_\lambda = \mathbb{E}_\lambda[g(Y)]$, and specified discrepancy measure $d$.

For instance, for target mean $E^*$ and variance $V^*$:

$$
\begin{aligned}
d &:= (E^* - \mathbb{E}[Y])^2 + (V^* - (\mathbb{E}[Y^2] - \mathbb{E}[Y]^2))^2, \\
g(Y) &= (Y, Y^2), \\
\hat{T}(E_1, E_2) &= (E_1, E_2 - E_1^2)
\end{aligned}
\tag{89}
$$

The idea of optimization is: iteratively, draw samples from the PPD to estimate the virtual statistics $\hat{T}_\lambda$, and solving for $\lambda$ by optimizing Equation (88).

To approximate the PPD moments, use Monte Carlo estimation and stochastic gradient descent, with reparameterization gradients (or REINFORCE), assuming that both $d(.)$ and $\hat{T}_\lambda$ are differentiable.

For PMF, they have:

$$
\begin{aligned}
\mathbb{E}_\lambda[Y_{ij}] &\approx \frac{1}{S_\theta S_\beta} \sum_{\epsilon_\theta \sim p_0} \sum_{\epsilon_\beta \sim p_0} \mathbb{E}[Y \mid \theta(\epsilon_\theta, \lambda)^T \beta(\epsilon_\beta, \lambda)], \\
\mathbb{E}_\lambda[Y_{ij} \mid \theta^T \beta] &\approx \frac{1}{C} \sum_{y \sim \mathrm{Poisson}(\theta^T \beta)} y.
\end{aligned}
\tag{90}
$$

For a given dataset, estimate $E^*$ and $V^*$ from the data, then optimize Equation (88) iteratively using GSD, where $d(.,.)$ is defined in Equation (89) and the moments in Equation (90).

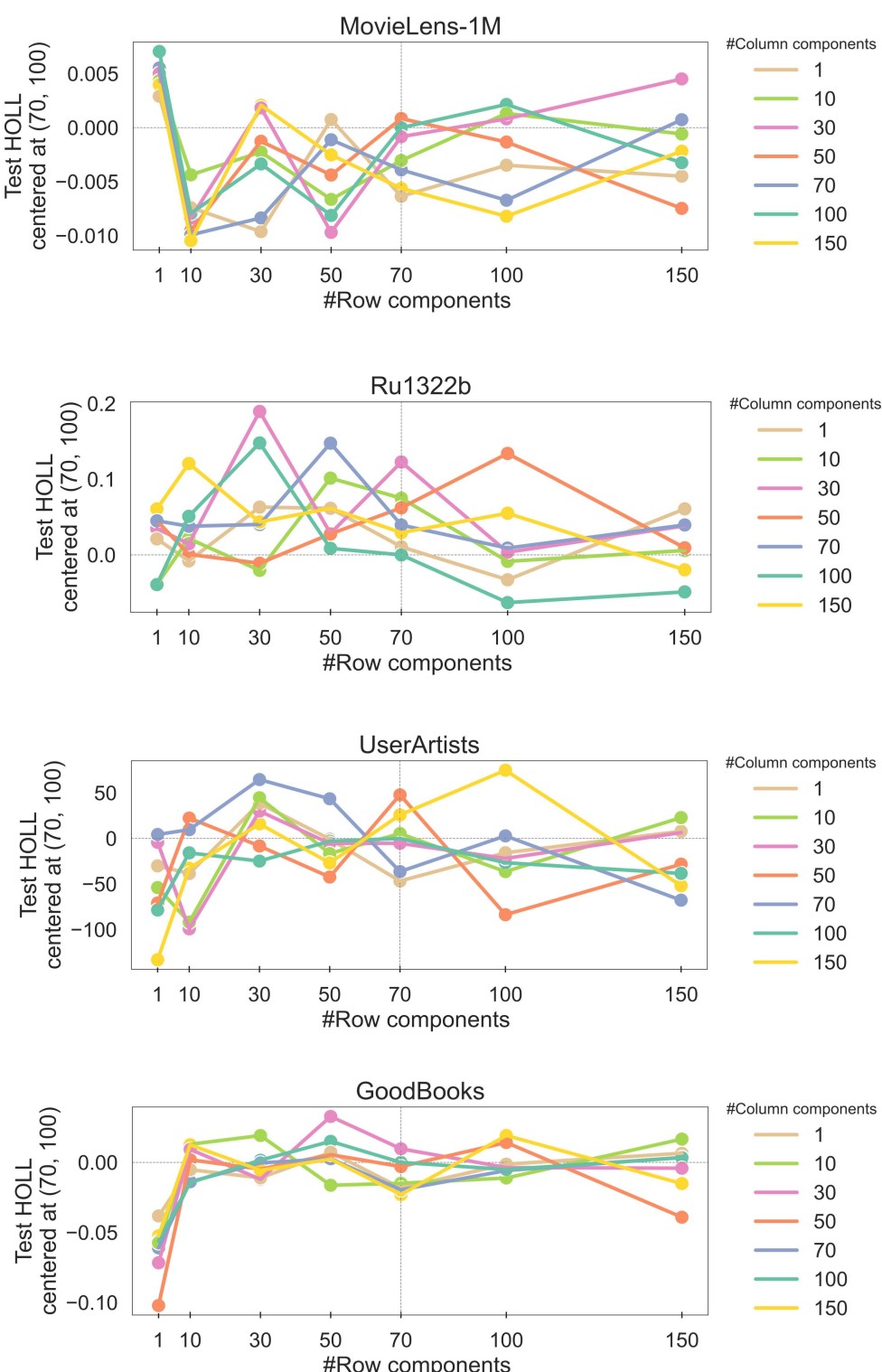

**Figure 8:** TwinEB is robust to the number of mixture components on real-world datasets. Test hold-out likelihood as a function of $K_r$, the number of row components (x-axis) and $K_c$, the number of column components (hue) for TwinEB with the Poisson likelihood. Each experiment is from 10 seeds. The test hold-out likelihood is centered at the result of $(K_r, K_c) = (70, 100)$, highlighted by the gray dashed lines. Note that the range of the vertical axis is orders of magnitude smaller than the range of the real world experiments, suggesting that TwinEB is not sensitive to the choice of the number of mixture components.

