# OpenReview forum: "Population Priors for Matrix Factorization"
_TMLR — Accepted by TMLR_

### Review · Reviewer_YEoN · 2024-08-03

**Summary Of Contributions:**

The paper introduces an empirical Bayes prior over factor matrices for matrix factorization, specifically Poisson and Gaussian matrix factorization. These priors are modeled as mixture distributions. Inference is performed using a variational objective. Conveniently, the ELBO allows for optimization of both the variational parameters of the factor matrices' posterior, as well as the parameters of the priors' mixture distribution. The method is tested on held-out log-likelihood in comparison to similar methods on standard benchmark data. It performs on par, sometimes favorably, sometimes worse than existing methods.

**Audience:**

Yes

**Claims And Evidence:**

Yes

**Requested Changes:**

- The manuscript needs to be proofread for comma punctuations. E.g. "Here, U and V are [...] latent vectors."

- In Fig. 4, the held-out performance increases with the number of components. What is the limiting behavior when using even more components? Does it match your expectations?

- How do you decide on the number of mixture components in your main experiments? Does it matter?

- In the discussion, I would like to see more of a critical assessment where this method fits in or improves over existing work.

- I may have missed it, but why can you not optimize row and column priors simultaneously (or rather iteratively in the same loop), instead of holding one fixed?


## Detailed Comments


### Introduction
I find the formulation in the introduction somewhat confusing. E.g. you say "Here \mathbf{U}_i and \mathbf{V}_j are per row and per column specific latent vectors". I get what this means from the context, the sentence itself is unclear. Similarly, you say "An observed matrix of data X then defined a posterior distribution p(U, V|X) over the row variables and the column variables." What are the "row and column variables"? Why not just say "... over the latent factor matrices" or so. Am I missing some subtlety here?

"It is applicable for priors over variables that have repeated draws from them" -> Sentence is unclear.

"The data that informs us about them is overlapping" -> What does this mean?

- [FYI] Another example of empirically set matrix factorization population priors: https://proceedings.mlr.press/v70/rukat17a/rukat17a


### 2 Related work
You say that mixture priors are more expressive than hierarchical priors. This is not generally true.

### 3 Empirical Bayes
What is P*? You're saying the "true (unobserved) distribution of observations". That seems nonsensical or, at least, requires some explanation.

I don't understand the point of section 3.1. Could you explain why we need this explanation?

You say "there is no distinction between local and global random variables" - I think you want to say "latent" instead of "random".

Why do you choose a log normal?

### 4 Experiments
"We studied Algorithm 1 ..." -> You probably want to write this and what follows in the present tense.

The train/test split procedure is confusing. Can you please explain and motivate the choices more clearly?

Sections 4.2 and 4.3 are named "Results", but also explain the experimental procedure.

### 5 Discussion

The last sentence is not complete.

**Strengths And Weaknesses:**

(+) Great to see the explicit comparison to da Silva et al and to Wang and Stevens.
(+) The approach is elegant, re-using the ELBO for prior and posterior optimization.
(-) Empirical results are rather weak. For some examples other methods perform better (Fig. 2). The authors downplay this as "comparable".
(-) Some design choices could be better-justified (see requested changes)
(-) Limited methodological novely

---

> ### Author Response · Authors · 2024-10-10
> **Rebuttal**
>
> We thank the reviewer for the time and energy spent on our manuscript. Please see our point by point response below.
>
> ## Requested changes
> 1. **The manuscript needs to be proofread for comma punctuations. E.g. "Here, U and V are [...]"**
>
> Thank you, we have proofread and updated the paper with special attention paid to the commas. E.g.:  *"Here, U and V, … Here, we use gradient ascent … Here, we examine … Next, we focus on… Here, we set …"*
>
> 2. **In Fig. 4, the held-out performance increases with the number of components. What is the limiting behavior when using even more components? Does it match your expectations?**
>
> These are great questions, thank you. With more components, the performance plateaus. We regenerated and updated Figure 4 to include more components up to 120.
>
> Please find the updated figure here: [anonymized link](https://figshare.com/s/f426e0a9985af6513c09).
> Caption: *TwinEB is robust to the number of mixture components in simulated studies. Test hold-out likelihood as a function of $K_r$, the number of row components (x-axis) and $K_c$, the number of column components (hue) for TwinEB with the Poisson likelihood. Each experiment is from 10 seeds.*
>
> We find that after an initial increase (about #mixtures = 50), the performance plateaus. We expected this behavior, which is when the mixture becomes flexible enough to approximate the empirical Bayes prior.
>
> 3. **How do you decide on the number of mixture components in your main experiments? Does it matter?**
>
> We arbitrarily set-up the number of mixture to be 70 for rows and 100 columns, which we found to be good enough.
> In order to motivate that choice, we now have conducted and included experiments on all four real-world datasets where we vary the number of row and column components from 1 to 150 components. The figure below shows the results. The results suggest that TwinEB is robust to the number of mixture components. We include it in Supplementary.
>
> Find the new figures here: [anonymized link](https://figshare.com/s/c2d6f745a9e1e1759082).
> Caption: *TwinEB is robust to the number of mixture components on real-world datasets. Test hold-out likelihood as a function of $K_r$, the number of row components (x-axis) and $K_c$, the number of column components (hue) for TwinEB with the Poisson likelihood. Each experiment is from 10 seeds. The test hold-out likelihood is centered at the result of ($K_r$, $K_c$) = (70, 100), highlighted by the gray dashed lines. Note that the range of the vertical axis is orders of magnitude smaller than the range of the real world experiment (c.f. manuscript Figure 3), suggesting that TwinEB is not sensitive to the choice of the number of mixture components.*
>
>
> 4. **In the discussion, I would like to see more of a critical assessment where this method fits in or improves over existing work.**
>
> Thank you for pointing this out. Taking your comment, and that of the other reviewers, into account, we have expanded the discussion section to better situate our method in the field, including covering more methods. We have added those paragraphs:
>
> - *Our method uses the widely applicable black box variational inference (BBVI); the availability and ease of use of automatic differentiation, our implementation brings flexible priors in the context of matrix factorization to a wider audience. For example, we studied our proposed method works on multiple data types, both Gaussian and Poisson likelihoods, and it is easy to extend it to other reparameterizable likelihoods. In contrast, the methods of Wang and Stephens and that of da Silva et al are tailored for only Gaussian and only count data only.*
>
> - *With TwinEB, the latent dimensions $L$ of the row-and column-wise variables is a hyperparameter. In practice, we found that setting $L$ at about 15 was generally good enough (used across all our experiments) and ablation studies in Figure 5 (Appendix) showed that the results were stable with the choice of $L$ ([GMF anonymized link](https://figshare.com/s/b9842221e90680a0c1e0), [PMF anonymized link](https://figshare.com/s/f513b0bd64790b9470ba)) However, an attractive property of the method of da Silva et al is that it can automatically set this value in closed form. Methods based on prior predictive statistics suggested by da Silva et al. (2023) could optimize the selection of L.*
>
> - *A limitation of our model is that it does not leverage auxiliary information. Auxiliary information is information in addition to the observed interaction matrix. For instance, for movie ratings, the additional information could be user-specific attributes including age, race, and occupation, while item-specific attributes could include description of the items. Methods like Wang et al (2024) [1]  and Van Linh et al, (2020) [2] propose ways to integrate auxiliary information to improve the inference in matrix factorization.*
>
> (please see the rest of rebuttal below)

---

> ### Author Response · Authors · 2024-10-10
> **Rebuttal (continue)**
>
> - *We use a linear function to model the dependence of the observed interactions on the per-row per-column latent variables. This has the advantage of being interpretable and prevent overfitting (e.g., in the analysis of single cell gene expression data [3]). However, in the presence of non-linear interactions, this model would be misspecified.  In that case, the link function could be changed, for e.g., Zhou et al. (2020) [4] use a multi-layered perceptron (MLP) to model the dependence of the observed interactions on the per-row per-column latent variables.*
>
> - *Finally, other flexible distribution learning methods can be used to specify expressive priors, such as normalizing flows used in Zhou et al. (2020), and one of our baseline methods, TwinNF.*
>
> [1] Wang, Zhiwei, et al. "MFAI: A Scalable Bayesian Matrix Factorization Approach to Leveraging Auxiliary Information." Journal of Computational and Graphical Statistics (2024): 1-11.
>
> [2] Van Linh, Ngo, et al. "Neural poisson factorization." IEEE Access 8 (2020): 106395-106407.
>
> [3] Svensson, Valentine, et al. "Interpretable factor models of single-cell RNA-seq via variational autoencoders." Bioinformatics 36.11 (2020): 3418-3421.
>
> [4] Zhou, Fan, et al. "Recommendation via collaborative autoregressive flows." Neural Networks 126 (2020): 52-64.”
>
> 4. **I may have missed it, but why can you not optimize row and column priors simultaneously (or rather iteratively in the same loop), instead of holding one fixed?**
>
> This is exactly what we are doing, we are not holding any of them fixed. The parameters for the row prior, the column prior and the variational posterior are all simultaneously updated (e.g., see Algorithm 1, third to last line).
>
>
> ## Detailed Comments
>
> 1. **I find the formulation in the introduction somewhat confusing. E.g. you say "Here \mathbf{U}_i and \mathbf{V}_j are per row and per column specific latent vectors". I get what this means from the context, the sentence itself is unclear. [...]**
>
>
> $\mathbf{U}\_{i}$ and $\mathbf{V}_{j}$ are mathematically defined in the previous sentence. We have updated the descriptive sentence into:
>
> *"$\mathbf{U}\_{i}$ is a latent variable representing row $i$, and $\mathbf{V}\_{j}$ is a latent variable representing column $j$. The variables have respective priors $P_{\theta_{\text{row}}}$ and $P_{\theta_{\text{col}}}$, each with hyperparameter $\theta_{\text{row}}$ and $\theta_{\text{col}}$."*
>
>
>
> 2. **"It is applicable for priors over variables that have repeated draws from them" -> Sentence is unclear. "The data that informs us about them is overlapping" -> What does this mean?**
>
> We believe those two sentences are very important and highlight the subtlety of empirical Bayes for matrix factorization. We thank you for asking for more precision. Here are the new versions of the sentences:
>
> - *"The EB idea is applicable to any prior distribution from which are drawn multiple IID variables. For example, all the $\mathbf{U}\_i$ are independently drawn from the same $P\_{\theta_{\text{row}}}$ and the $\mathbf{V}\_j$ are independently drawn from the same $P_{\theta_{\text{col}}}$."*
>
> - *"the same data, namely contains information about them, namely the observed data $\mathbf{X}_{i,j}$ contains information about both $\mathbf{U}_i$ and $\mathbf{V}_j$."*
>
>
> 3. **[FYI] Another example of empirically set matrix factorization population priors: [https://proceedings.mlr.press/v70/rukat17a](https://proceedings.mlr.press/v70/rukat17a)**
> Thank you for pointing out this interesting work, we now mention it in the related work. We point out that in this work, the use of terminology "Empirical Bayes" refers to selecting a single and fixed values for the prior hyperparameter, whereas the novel methodology of our paper explains how to let the Empirical Bayes methodology learn automatically the prior hyperparameters (via gradient updates).
>
> [1] Rukat, Tammo, et al. "Bayesian boolean matrix factorisation." International conference on machine learning. PMLR, 2017.
>
> 4. **You say that mixture priors are more expressive than hierarchical priors. This is not generally true.**
>
> Thank you for this astute observation. We have clarified the sentence and use the expression:
> *"Our work involves learnable priors"* instead of *... more expressive*.

---

> ### Author Response · Authors · 2024-10-10
> **Rebuttal (continue)**
>
> 2. **"It is applicable for priors over variables that have repeated draws from them" -> Sentence is unclear. "The data that informs us about them is overlapping" -> What does this mean?**
>
> We believe those two sentences are very important and highlight the subtlety of empirical Bayes for matrix factorization. We thank you for asking for more precision. Here are the new versions of the sentences:
>
> - *"The EB idea is applicable to any prior distribution from which are drawn multiple iid variables. For example, all the $\mathbf{U}\_i$ are independently drawn from the same $P\_{\theta_{row}}$ and the $\mathbf{V}\_j$ are independently drawn from the same $P_{\theta_{col}}$."*
>
> - *"the same data, namely contains information about them, namely the observed data $\mathbf{X}_{i,j}$ contains information about both $\mathbf{U}_i$ and $\mathbf{V}_j$."*
>
>
> 3. **[FYI] Another example of empirically set matrix factorization population priors: [https://proceedings.mlr.press/v70/rukat17a](https://proceedings.mlr.press/v70/rukat17a)**
> Thank you for pointing out this interesting work, we now mention it in the related work. We point out that in this work, the use of terminology "Empirical Bayes" refers to selecting a single and fixed values for the prior hyperparameter, whereas the novel methodology of our paper explains how to let the Empirical Bayes methodology learn automatically the prior hyperparameters (via gradient updates).
>
> [1] Rukat, Tammo, et al. "Bayesian boolean matrix factorisation." International conference on machine learning. PMLR, 2017.
>
> 4. **You say that mixture priors are more expressive than hierarchical priors. This is not generally true.**
> Thank you for this astute observation. We have clarified the sentence and use the expression:
> *"Our work involves learnable priors"* instead of *... more expressive*.
>
> 5. **What is P\*? You're saying the "true (unobserved) distribution of observations". That seems nonsensical or, at least, requires some explanation. I don't understand the point of section 3.1. Could you explain why we need this explanation?**
>
> Thank you for the question, we agree that it can benefit from more explanation, but it is sensical, in fact, it is even key to explaining the empirical Bayes (EB) methodology.
>
> In the paper, we start by assuming that the data $X$ comes from a distribution $P^*(X)$, which is unknown, for instance, we are thinking of biochemical processes in cells that give rise to the observed gene expression profiles. Then, given a model class $P_\theta(X | z)$, the goal of EB is to find the prior $\pi(z)$ over $z$ that will match the population distribution $P^*(X) = \int_{z} \pi(z)P_\theta(X|z)$. We have rephrased the paragraph to distinguish between the true population distribution $P^*(X)$ and the model-based distribution $P_{\theta, \pi}(X)$ that our model TwinEB finds.
>
> We use section 3.1 to introduce the concept of population priors in a simpler setting than matrix factorization, where there is a single prior to be set. We acknowledge that it introduced redundancy; however, in previous expositions, some readers reported that this introductory section had helped with understanding section 3.2.
>
> 6. **Why do you choose a log normal?**
>
> We use a Poisson response function $P(X_{i,j} | U_i^\top V_j)$ which requires $U_i^\top V_j$ to be positive real. Hence, the priors over $U_i$ and $V_j$ should be over positive real numbers. We chose to parametrize such priors using log-normal distributions because they are over positive reals, they have a simple parametrization and they are numerically stable. But other distributions can be used too. The log-normal distribution in our experiments yielded the most stable inference. The variational family can be specified in our code base using the option --var_family.
>
> 8. **"We studied Algorithm 1 ..." -> You probably want to write this and what follows in the present tense.**
>
> Thank you for pointing this out, we updated the text.

---

> > ### Author Response · Authors · 2024-10-10
> > **Rebuttal (continue)**
> >
> > 9. **The train/test split procedure is confusing. Can you please explain and motivate the choices more clearly?**
> >
> > Thank you for pointing this out. We have clarified the text in section “4.2 Evaluation Metric and Baselines”, to better motivate our choices. We have also moved some detail to the supplement to reduce clout.
> >
> > *“We evaluate model performance using the likelihood of unseen interactions. To build intuition, we will use the example of user-item interactions where rows indicate users and columns items, and the entries of the input matrix their observed interactions. One approach to testing performance is to hold-out a fraction of the entries of the input matrix, and to train the model on the held-in portion, and test performance on the held-out entries. In this approach, all users are observed during training. To test generalizability to cases when new users are observed, we can further hold-out a portion of users (rows) during training, and then test the performance on these hold-out users. Specifically, we will test the performance on masked entries of these hold-out users. This procedure measures strong generalization (Steck, 2019).*
> >
> > *At test time, we need to learn user-specific latents for the held-out rows. For this, we run the model in a special training step where the column latents are held fixed. Since these held-out users are nevertheless coming from the same pool as our training users, we use only 30% of the observed entries for this special training step. This may result in a more accurate measurement of out-of-distribution generalizability by breaking spurious correlations between items and users due to biases present in the current dataset. These biases may include the time-period of data collection, platforms used that may not exist in a new dataset. Finally, we report the model performance on 30% of the unseen entries of the held-out users.”*
> >
> > 10. **The last sentence is not complete.**
> >
> > It is actually complete, the "each" refers to the variables in the previous sentences. We updated the sentence for clarity.

---

### Review · Reviewer_zHrL · 2024-08-09

**Summary Of Contributions:**

This paper proposes a new method for simultaneous probabilistic matrix factorization and prior estimation by maximizing ELBO with respect to both posterior distribution and prior parameters. The author claims that the algorithm performs better than manually tuning the hyper-parameters for the prior or selecting priors with Empirical Bayes.

**Audience:**

Yes

**Broader Impact Concerns:**

None.

**Claims And Evidence:**

No

**Requested Changes:**

These changes are critical to securing the recommendation for acceptance. More details and numerical experiments are needed to support the advance of this method.
1.  There should be more comparison with other methods in simulation and on real datasets, including Wang & Stephens (2021), da Silva et al. (2023) that demonstrates the strengths and weakness of each method. Now the comparison with Wang & Stephens (2021) is concluded with one sentence "flashr crashes on all but the least sparse dataset, Ru1322b." More insights are needed.
2. The selection of the real datasets needs to be reconsidered. All real datasets have less than 20% of non-zero elements. Why apply Gaussian models to them (Section 4.3)?
3.  The methods under comparison have not been clearly described. Is 'row prior variance' the hyper-parameter for GMF? How is it used in GMF?
4. "We note that it is not straightforward to compare our method to that of Jiang & Zhang (2009);" Is Jiang & Zhang (2009) a method for matrix factorization?

These changes will strengthen the work in my view:
1. The description of "population prior" is too lengthy on pages 2-4.
2. Abbreviations appear without definition, such as HOLL.

**Strengths And Weaknesses:**

Strengths: This seems to be a novel approach, and the no-tuning feature is appealing.

Weakness:  Lack of scope and clarity in numerical studies, so it is unclear how much advancement this method brings 'in performance' compares to existing approaches.

---

> ### Author Response · Authors · 2024-10-10
> **Rebuttal**
>
> We thank the reviewer for the time and energy spent on our manuscript. Please see our point by point response below.
>
>
> ## Requested Changes
>
> 1. **There should be more comparison with other methods in simulation and on real datasets, including Wang & Stephens (2021), da Silva et al. (2023) that demonstrates the strengths and weakness of each method. Now the comparison with Wang & Stephens (2021) is concluded with one sentence "flashr crashes on all but the least sparse dataset, Ru1322b." More insights are needed.**
>
> Thank you for proposing improvements to the paper. We note that we compare our method to that of Wang & Stephens (2021) and da Silva et al. (2023) on all our real datasets. In addition, we ran both methods on our simulated data. We found that, in our new simulated studies, Wang & Stephens performed on par with Twin EB, with TwinEB doing slighly better. The method of da Silva et al. (2023) did not yield admissible parameters on the simulated dataset; please see our description below for our hypothesis.
>
> | **Method**                 | **Mean Absolute Error** |
> |----------------------------|------------------------|
> | Wang & Stephens (2021)     | 0.37                   |
> | TwinEB (GMF)               | 0.35                   |
>
>
> In addition, we have added comparisons to two other baseline methods:
> - one is based on normalizing flows, we describe it in response to reviewer XJwX27 (Point 3)
> - the other is based on learning the hyper-parameters of Gamma priors for our PMF experiments (close to Hierarchical Poisson Factorization of Gopalan et al., 2015 ), we detail it below.
>
> In the new baseline, we infered the hyper-parameters of the two Gamma priors. This represents a more exhaustive search in the hyper-parameter space of a model with uni-modal priors. We will add these as reference points to Figure 3. We note that in the Ru1322b dataset this method has the worst performance (as measured by Test Hold-Out likelihood), suggesting that the uni-modal prior is not well specified for that dataset.
>
> | **Dataset**      | **Test HOLL** |
> |------------------|---------------|
> | UserArtists      | -1692.74      |
> | Ru1322b          | -10.00        |
> | MovieLens 1M     | -2.43         |
> | GoodBooks        | -3.94         |
>
>
> We do compare with the method of da Silva et al. As we report in Table 1, for two of our real-world datasets, we report results from this method. In the table below, we report the values of the hyperparameter that it infers for these two datasets:
>
> | **Dataset** / **Hyperparameters**     | **Row Shape (a)** | **Row Rate (b)** | **Column Shape (c\)** | **Column Rate (d)** |
> |------------------|-------------------|------------------|---------------------|--------------------|
> | **MovieLens-1M** | -0.60             | -0.60            | -0.21               | 10.15              |
> | **GoodBooks**    | -0.82             | -0.82            | -0.01               | 30.48              |
>
>
> In various occasions, this procedure provides negative values for the shape and rate hyperparameters of the prior, however, the hyperparameters of a Gamma distribution need to be positive.
>
> We will add a more detailed summary of this method to our supplement. Briefly:
>
> - *da Silva et al. (2022) find the prior hyper-parameters for the prior, such that the (moments) of the marginal distribution of the observations under the model, match the (empirical moments of the) observations well. While we are not certain why the method fails when it does, we have noticed that it tends to fail on sparse datasets. One potential pitfall is that their method relies on computing the empirical correlation for the rows and columns, and this appears to be unstable in practice. We have used their original codebase as well as our own reimplementation, but have not been able to circumvent this problem.*
>
> The method of Wang & Stephens (2021), does not immediately lend itself to holdout-likelihood based comparison. However, we have used matrix completion as a procedure to compare their method to TwinEB (ours). We report this at the end of section 4.3 in the manuscript: On the single cell gene-expression dataset (Ru1322b),  we achieve a mean absolute error of  0.51 to flashr’s 0.64.
>
> It is not clear to us why the method of Wang & Stephens (2021) does not run on our other datasets. We will update the wording to:
> _ *The optimization objective encounters NaN values, which halt execution and terminate without results.*
>
> We will add a more detailed review of their method to our supplementary information. It would be an interesting direction to debug the flashr’s code base, however, we feel it is out of the scope of our current manuscript. We hope that adding the detailed review of their method will help inform the readership.

---

> > ### Author Response · Authors · 2024-10-10
> > **Rebuttal (continue)**
> >
> > 2. **The selection of the real datasets needs to be reconsidered. All real datasets have less than 20% of non-zero elements. Why apply Gaussian models to them (Section 4.3)?**
> >
> > Thank you for the question. Indeed the real datasets we analyze are sparse. There is a history of applying gaussian models to sparse datasets, even exactly those used in our paper (MovieLens, gene expression) [1]. For example, sparse count data is routinely normalized and modeled as Gaussian data in single-cell genomics.
> >
> >  [1] Wang, Zhiwei, et al. "MFAI: A Scalable Bayesian Matrix Factorization Approach to Leveraging Auxiliary Information." Journal of Computational and Graphical Statistics (2024): 1-11.
> >
> > 3. **The methods under comparison have not been clearly described. Is 'row prior variance' the hyper-parameter for GMF? How is it used in GMF?**
> >
> > The explanation of "row prior variance" for GMF is detailed in "section 4.3 - Results: Gaussian Matrix Factorization". We have emphasized the text with an example:
> > *"For example, in Figure 2, the left column of plots corresponds to fixing the prior on the column-wise variables to $P(\boldsymbol{V}_j) = \mathcal{N}(0, 1)$ and varying the prior on the row-wise variables as $P(\boldsymbol{U}_i) = \mathcal{N}(0, \sigma^2)$ with $\sigma^2  \in \{0.001, 0.01, 0.1, 1, 10\}$."*
> >
> > 4. **"We note that it is not straightforward to compare our method to that of Jiang & Zhang (2009);" Is Jiang & Zhang (2009) a method for matrix factorization?**
> >
> > Thank you for pointing this out. The citation should have been to *Zhong et al., 2022*, which we introduced in Section 2. We have updated the text to reflect this.
> >
> > 5. **The description of "population prior" is too lengthy on pages 2-4.**
> >
> > Thank you for the comment. We have clarified and shortened this section.
> >
> > 6. **Abbreviations appear without definition, such as HOLL.**
> >
> > Thank you for noticing this. HOLL stands for held-out likelihood, that we had defined in section 4.2. (Eq. 26). We have now added a clarifying statement as follows: *“For each model, we report the test held-out likelihood (test-HOLL) of the random seed that achieved the best validation HOLL.”*

---

### Review · Reviewer_XJwX · 2024-09-28

**Summary Of Contributions:**

This paper introduces **twin population priors**, a novel approach for determining priors in **Bayesian Matrix Factorization (MF)** within an **Empirical Bayes (EB)** framework. Traditional methods for setting priors often rely on fixed parametric distributions (e.g., Gaussian, Gamma) or cross-validation, which can be computationally expensive and may not capture the data's complexity effectively. To address this, the authors propose data-driven priors for both row and column latent variables in the matrix, which are learned directly from the observed data.

The main contributions of the paper include:

1. **Twin Population Priors**: A framework that separately learns empirical Bayes priors for both row and column latent variables, aligning the model’s marginal distributions with the true population distribution.

2. **Mixture Distributions for Flexibility**: Both row and column priors are modeled as mixtures of parametric distributions, offering a flexible prior that can capture more intricate patterns in the data.

3. **Joint Optimization via Variational Inference**: The authors develop a variational inference algorithm that jointly optimizes the twin population priors and the posterior distributions. This approach allows efficient learning of both the priors and the latent variables.

4. **Comprehensive Empirical Evaluation**: The method is evaluated on multiple real-world datasets, including movie ratings and gene expression data, demonstrating improved performance over traditional parametric priors and comparable EB methods. Particularly, the twin population priors show robustness in sparse data scenarios.

**Audience:**

Yes

**Broader Impact Concerns:**

The paper should mention potential ethical concerns in the context of recommender systems applications. Specifically, empirical priors derived from the data may inherit and amplify biases present in the data, which could lead to biased outcomes in real-world applications. Including a discussion of these potential biases and suggesting mitigation strategies (e.g., fairness-aware priors) would be a valuable addition to the broader impact section.

**Claims And Evidence:**

Yes

**Requested Changes:**

1. **Critical**:
   - **Clarify Notation**: The notational ambiguities should be addressed to make the paper more accessible. The distinction between hyperparameters, random variables, and distributions must be clearly defined. Adopting more standard Bayesian notations would enhance the clarity and understanding of the model’s structure. A section in the background section or introduction clearly defining the notation used is important in this case (for example, observed variables, latent variables, hyperparameters, pdf/cdf, sampling distribution, empirical distributions).

2. **Important**:
   - **Expand the Discussion on Flexible Priors**: The literature review and discussion should incorporate a broader range of methods for flexible prior specification, including models that leverage side information, neural networks, and decision trees. This would give the reader a more holistic view of the field and place the contribution in the context of existing techniques.

3. **Important**:
   - **Comparison with Flow-Based Models**: The authors should include a comparison with flow-based methods, such as CAF, which offer flexible ways to approximate complex posterior distributions. This would provide valuable insights into the strengths and limitations of the variational inference approach used in the paper. Discussing the differences and limitations is relevant if such a comparison is impossible.

4. **Important**:
   - **Justification for Latent Dimensionality**: The paper should provide a more detailed explanation of how the latent dimensionality $ L $ was chosen. Incorporating prior predictive methods for optimizing $ L $ would add methodological rigor and strengthen the empirical results.

5. **Important**
- **Theoretical Discussion on Prior Predictive Properties**: A theoretical discussion on the prior predictive properties of the twin population priors would significantly strengthen the paper’s theoretical foundation. This could include calculations based on mixture distributions, as seen in prior work (e.g., Silva et al., 2023) combined with known moments for mixtures (https://math.stackexchange.com/questions/195911/calculation-of-the-covariance-of-gaussian-mixtures), and would also provide an opportunity to introduce a new baseline—**twin priors with moment matching**. Such a discussion would offer insights into the behavior of the priors before data is observed, which is crucial for understanding their impact on model performance.

**Strengths And Weaknesses:**

### **Strengths**

1. **Novelty**: The introduction of twin population priors is a novel contribution that provides a principled solution to the problem of prior selection in Bayesian MF. The concept of empirically learning priors while leveraging the natural separation between rows and columns in a matrix is intuitive and effective.

2. **Flexibility via Mixture Distributions**: By using mixture distributions for priors, the model can capture a wide range of data distributions, making it highly adaptable to various types of data and patterns.

3. **Simultaneous Prior and Posterior Learning**: The variational inference approach allows for simultaneous learning of priors and posterior distributions, which reduces the need for manual tuning of hyperparameters and makes the method more robust.

4. **Strong Empirical Results**: The authors provide convincing empirical evidence across different datasets, showing that their method outperforms traditional manually tuned priors and other empirical Bayes approaches. This is particularly relevant in scenarios with sparse data, such as movie recommendation systems, where the method demonstrates superior predictive accuracy.

---

### **Weaknesses**

1. **Notational Inconsistencies**: The paper suffers from notational ambiguities, particularly in using $ \theta_r $ and $ \theta_c $, which denote distributions and latent variables. A clearer distinction between hyperparameters, distributions, and latent variables is necessary. For instance, using $ \phi $ for hyperparameters and reserving $ \theta $ for latent variables would improve readability.
   - In equations like $ U_i \sim P_{\theta_r}(U) $ and  $ V_j \sim P_{\theta_c}(V) $, the parameters $ \theta_r $ and $ \theta_c $ are used both as the distributions' parameters and the latent variables' labels, which creates confusion. A more typical approach would be to separate hyperparameters and variables (e.g., using different symbols or subscripts).
   - In other places, such as the factorization of the joint distribution (Equation 4), using $ \theta $ as global and local latent variables is problematic. This contradicts typical Bayesian literature conventions, which recommend distinguishing hyperparameters (e.g., $ \phi $) from random variables (e.g., $ z $).
   - Potentially avoid using subscripts to introduce variables rather than using them to index variables' dimensions. The separation between random variables as conditional and hyperparameters could be achieved by using `|` and `;.`

2. **Limited Comparison with Other Flexible Priors**: The paper does not sufficiently compare its method with other approaches incorporating flexible priors through side information, neural networks, or decision trees. Several methods, such as **Neural Poisson Factorization** (Ngo et al., 2020) and **MFAI: A Scalable Bayesian Matrix Factorization Approach** (Wang et al., 2024), use advanced techniques like neural networks and gradient-boosted trees to enhance prior specification. A discussion or comparison with these methods would provide a more comprehensive evaluation of the method’s strengths and weaknesses.

3. **Absence of Comparison with Flow-Based Methods**: The paper does not explore flow-based methods, such as **Collaborative Autoregressive Flows (CAF)** (Zhou et al., 2020), which could offer a more flexible way of capturing complex posterior distributions. While the authors highlight the advantages of EB approach, a comparison with flow-based models that represent non-linear user-item interactions would provide a deeper understanding of the trade-offs involved.

4. **Explanation of Latent Dimensionality Selection**: The rationale behind selecting the latent dimensionality $ L $ is not clearly explained. It would be useful for the authors to discuss how the dimensionality was chosen and whether methods like prior predictive statistics suggested by da Silva et al. (2023) could optimize the selection of $ L $. This would provide further rigor to the experimental setup.

5. **No theoretical insight on the role of number mixture components**: some theoretical insight about the role the number of mixture components plays in properties of the posterior or prior predictive distribution could help understand the choice of such parameters.

---

> ### Author Response · Authors · 2024-10-10
> **Rebuttal**
>
> We thank the reviewer for the time and energy spent on our manuscript. Please see our point by point response below.
>
> 1. **Critical: Clarify Notation: The notational ambiguities should be addressed to make the paper more accessible. The distinction between hyperparameters, random variables, and distributions must be clearly defined. [...]**
>
> Thank you for the comment and suggestion, we have updated and simplified the notations and it has greatly improved the paper's clarity.
>
> - Priors on latent variables were written $P_\pi(z), P_U(V),$ and $P_V(U)$. We now write them as $\pi(z)$, $\pi_{row}(U)$ and $\pi_{col}(V)$ as we believe it will help avoid confusion between priors and hyperparameters.
> - We removed the dependence of the response distribution on $\theta$ from in section 3.1 and eq. 5. We were not using this variable, and it was incorrectly presented as a hyperparameter *and* random variable. This further avoids confusion with the variational parameters $\theta^c, \theta^r$.
> - We renamed the mixture weights $(\pi_i)$ into $(w_i)$ to avoid overloading the notation $\pi$.
> - We uniformized how we refer to row- and column-specific variables by using the subscript XXX$_{\text{row}}$ instead of XXX$^{r}$.
>
> Overall, these changes result in a clearer presentation and a paper that is easier to understand, thank you for the suggestions. We also expanded the notation section in the background (section 3) as you suggested.
>
>
> 2. **Important: Expand the Discussion on Flexible Priors: The literature review and discussion should incorporate a broader range of methods for flexible prior specification, including models that leverage side information, neural networks, and decision trees. This would give the reader a more holistic view of the field and place the contribution in the context of existing techniques.**
>
> Thank you for your great recommendations. We have expanded our discussion, and put our method in context using the 3 methods mentioned above.
>
> The related works now includes:
> - *Wang et al (2024) [1] uses auxiliary information to improve inference in matrix factorization. They  assume the existence of an auxiliary matrix $G$, comprising per user extra features. Their method is closest to that of Wang & Stephenss, except that they define the per row latent variables as a function of the auxiliary matrix G. They use an ensemble of regression trees to model the functional dependence of the row-wise latents on the auxiliary variables. They use a Gaussian likelihood, with a mean as a linear function of the user and item representations.*
> - *Van Linh et al, (2020) [2] incorporate auxiliary information about the items (columns). They assume descriptions of each item exists. They use pre-computed word-embeddings to train a neural network to learn user representations. They use a Poisson likelihood with a rate computed as the linear combination of the user and item representations.*
> - *Finally, other flexible distribution learning methods can be used to specify expressive priors. For instance, Zhou et al. (2020) extend variational autoencoders (VAEs) and use normalizing flows to learn flexible priors on the latent embeddings of users and items. They combine user and item auxiliary information with these latent embeddings to learn user and item representation that are then passed through multi-layered perceptrons to generate the parameter of a Bernoulli distribution.*
>
> The discussion now includes:
> - *A limitation of our model is that it does not leverage auxiliary information. Auxiliary information is information in addition to the observed interaction matrix. For instance, for movie ratings, the additional information could be user-specific attributes including age, race, and occupation, while item-specific attributes could include description of the items. Methods like Wang et al (2024) [1]  and Van Linh et al, (2020) [2] propose ways to integrate auxiliary information to improve the inference in matrix factorization.*
>
> - *We use a linear function to model the dependence of the observed interactions on the per-row per-column latent variables. This has the advantage of being interpretable and prevent overfitting (e.g., in the analysis of single cell gene expression data [3]). However, in the presence of non-linear interactions, this model would be misspecified.  In that case, the link function could be changed, for e.g., Zhou et al. (2020) [4] use a multi-layered perceptron (MLP) to model the dependence of the observed interactions on the per-row per-column latent variables.*
>
> (please see the rest of the rebuttal below)

---

> ### Author Response · Authors · 2024-10-10
> **Rebuttal (continue)**
>
> [1] Wang, Zhiwei, et al. "MFAI: A Scalable Bayesian Matrix Factorization Approach to Leveraging Auxiliary Information." Journal of Computational and Graphical Statistics (2024): 1-11.
> [2] Van Linh, Ngo, et al. "Neural poisson factorization." IEEE Access 8 (2020): 106395-106407.
> [3] Svensson, Valentine, et al. "Interpretable factor models of single-cell RNA-seq via variational autoencoders." Bioinformatics 36.11 (2020): 3418-3421.
> [4] Zhou, Fan, et al. "Recommendation via collaborative autoregressive flows." Neural Networks 126 (2020): 52-64.”
>
>
>
> 3. **Important: Comparison with Flow-Based Models: The authors should include a comparison with flow-based methods, such as CAF [...]**
>
>
> Thank you for the suggestion. We have now expanded our discussion to mention flow based methods, specifically the method of Zhou et al., 2020 [4]. Below, we give a brief summary of the CAF method, compare it to TwinEB, and present results from an experiment using flow based priors in our framework.
>
> A simplified generative model for the CAF model:
> $$
> \begin{split}
> U_c, V_c, \epsilon &\sim N(0, 1) \\  % User and item auxiliary information, noise
> Z_U^{0} &\leftarrow f_{\theta} ( U^i, \epsilon) \\  % initial latent embedding per user
> Z_U^{K} &\leftarrow \text{AR}(Z_U^{0}; K)  \\ % apply normalizing flows
> Z_V^{0} &\leftarrow f_{\theta} ( V^i, \epsilon)  \\ % initial latent embedding per item
> Z_V^{K} &\leftarrow \text{AR}(V_U^{0}; K) \\   % apply normalizing flows
> U_i &\leftarrow g((Z_U^{K}) + U_c \\   % incorporate per user auxiliary information
> V_j &\leftarrow g(Z_V^{K}) + V_c  \\  % incorporate per item auxiliary information
> P(X_{i,j} | U_i, V_j) &= \text{Bern}(\log (\text{MLP} ( U_i, V_j ))) % the likelihood
> \end{split}
> $$
>
>
> AR refers to the auto-regressive flows, and MLP to multilayer perceptrons. Roughly speaking, CAF (Zhou et al, 2020) extends the framework of VAEs by passing the latent user and item embeddings ($Z_u$ and $Z_v$) from the encoders ($f(.)$) through a normalizing flow module with $K$ steps (AR), and then decode them (g) into user and item latent variables. Finally it uses a neural network (MLP) to link $U$ and $V$ to the observed interactions in a non-linear manner.
>
>
> We do not compute low-dimensional embeddings for our latent vectors, and we use a linear function to link the user and item latents with the observed interaction. As a result, a direct comparison to CAF is not feasible.  However, we have implemented a Normalizing Flow (NF) prior by stacking 10 Planar flows (similar to CAF) to define a flexible prior distribution. Each Planar flow applies an affine transformation to the latent variables $\mathbf{z}$ through the mapping:
> $$\mathbf{f}_z = \mathbf{z} + \mathbf{u} \cdot \tanh(\mathbf{w}^T \mathbf{z} + b),$$ where $\mathbf{u}$, $\mathbf{w}$ are learnable parameters and $b$ is a bias term. The determinant of the Jacobian is computed as: $$\text{det}(\nabla \mathbf{f}_z) = \left| 1 + \mathbf{u}^T \left( (1 - \tanh^2(\mathbf{w}^T \mathbf{z} + b)) \cdot \mathbf{w} \right) \right|. $$
>
> We do not claim that this implementation cannot be improved. Nevertheless, we find that TwinEB compares well with this NF prior, out-performing it in 3 out of the 4 real-world datasets. We show the result in the following table, where the second column shows the difference between Test-HOLL of TwinEB and NF (a positive value indicates that TwinEB performs better). We have used the same training procedure as in the main text, with 10 seeds.
>
> | **Dataset**      | **Delta-Test HOLL** |
> |------------------|----------------------------------------|
> | UserArtists      | 350.46                                 |
> | Ru1322b          | -0.12                                  |
> | MovieLens 1M     | 0.18                                   |
> | GoodBooks        | 0.40                                   |
>
> where  Delta-Test HOLL = Test HOLL\_TwinEB - Test HOLL\_NF.
>
> We add the following to our discussion:
>
> *By stacking multiple Planar Flows, Normalizing flow allows for more flexible and expressive transformations. The model begins with a simple base distribution and iteratively transforms it into a complex distribution that can capture more intricate patterns in the data.*

---

> > ### Author Response · Authors · 2024-10-10
> > **Rebuttal (continue)**
> >
> > 4. **Important: Justification for Latent Dimensionality: The paper should provide a more detailed explanation of how the latent dimensionality L was chosen. Incorporating prior predictive methods for optimizing L would add methodological rigor and strengthen the empirical results.**
> >
> > Ideally, the attractive feature of TwinEB is to not have to tune hyperparameters. We find that the results are robust with the choice of $L$.  We have expanded our discussion section to address this as a limitation of our method.
> >
> > *With TwinEB, the latent dimensions $L$ of the row-and column-wise variables is a hyperparameter. In practice, we found that setting $L$ at about 15 was generally good enough (used across all our experiments) and ablation studies in Figure 5 (Appendix) showed that the results were stable with the choice of $L$ ([GMF anonymized link](https://figshare.com/s/b9842221e90680a0c1e0), [PMF anonymized link](https://figshare.com/s/f513b0bd64790b9470ba)). However, an attractive property of the method of da Silva et al is that it can automatically set this value in closed form. Methods based on prior predictive statistics suggested by da Silva et al. (2023) could optimize the selection of L.*
> >
> >
> > 5. **Important: Theoretical Discussion on Prior Predictive Properties: A theoretical discussion on the prior predictive properties of the twin population priors would significantly strengthen the paper’s theoretical foundation. [...]**
> >
> > We appreciate your thoughtful recommendation. This is a great direction for our future research. We will expand the discussion to reflect this direction.
> >
> >
> > 6. **Broader Impact Concerns: The paper should mention potential ethical concerns in the context of recommender systems applications. Specifically, empirical priors derived from the data may inherit and amplify biases present in the data, which could lead to biased outcomes in real-world applications. [...]**
> >
> > Thank you for pointing out this potential ethical concern. We have updated the text to reflect this as follows:
> >
> > *As a data-driven approach, our method is subject to potential pitfalls inherent in such methods, including a tendency to inherit and amplify biases present in the data. If employed without proper supervision, this can lead to serious consequences including biased and/or discriminatory outcomes in real-world applications. These biases implicitly discriminate based on protected attributes (e.g., race, gender). One line of research assumes specific protected features are known, and enforces fairness by decoupling these features from learnt latent variables [1]. If fairness criteria can be formulated in terms of a regularization term, it can be reinterpreted as learning fairness-aware priors and incorporated in our framework [2].*
> >
> > [1] Togashi, Riku, and Kenshi Abe. "Fair matrix factorisation for large-scale recommender systems." arXiv preprint arXiv:2209.04394 (2022).
> >
> > [2] Zhu, Ziwei, Xia Hu, and James Caverlee. "Fairness-aware tensor-based recommendation." Proceedings of the 27th ACM international conference on information and knowledge management. 2018.

---

### Author Response · Authors · 2024-10-10
**Summary of our rebuttal**

Dear AE and reviewers,

We are grateful to you for handling and reviewing our paper. The reviews have helped us improve the manuscript.


We have submitted a point by point rebuttal to each reviewer's comments.
Please see below a summary of our paper, the reviews, and our responses.

Our paper derives empirical Bayes (EB) priors for probabilistic matrix factorization, derives a black box variational inference algorithm that can be easily generalized to different model specifications, and shows empirically on multiple real-world datasets, and in comparison to multiple baseline methods, that this strategy is effective.

The reviewers agreed that the approach is elegant and novel, but also cited some criticisms and offered ways to improve the paper:

- Clarification of the notations
- More details in our experimental results
- Comparison to, and discussion of additional relevant methods
- Exploration of the broader impacts

To address these points, we have significantly revised the manuscript:

- We added two new baseline methods
- We added details of the competing methods to the Appendix.
- We added multiple extra experiments, including the following:
    - Comparison to a baseline method based on normalizing flows
    - Comparison to a baseline method akin to hierarchical poisson factorization
    - Experiments with additional number of mixture components in our simulated studies
    - Experiments to measure the robustness of our method to the choice of number of mixture components on all real datasets
    - Experiments to measure the robustness of our method to the choice of the size of the row and column latents
    - Experiments to compare baselines with our method on simulated data
- We greatly expanded the discussion to better situate our method in the field.
- We overhauled our notations. In particular, there was a confusing sentence calling the hyperparameter $\theta$ as global latent variables. We have removed that sentence.

In light of these changes, we hope the AE and reviewers will feel that the paper is a worthy contribution to TMLR.

We are looking forward to more discussion with the reviewers.

Respectfully yours.

---

> ### Comment · Reviewer_XJwX · 2024-10-11
>
> Hi, thank you for the engaging discussion and work put into the paper. Based on the discussion, I see significant improvement. I will reply to each of your detailed comments individually, but I wanted to request an updated version of the entire manuscript to be uploaded. I prefer constructing my answer based on a holistic view of the improvements rather than partial incremental aspects. Thank you!

---

> > ### Author Response · Authors · 2024-10-12
> > **Updated revision**
> >
> > Hi,
> >
> > Thank you very much for taking the time to review our paper.
> >
> > We have uploaded a revised version of the manuscript.
> >
> > For your convenience, we have also uploaded a version of the main text with tracking enabled, so that the changes to the notation are clear. This document is uploaded as the compressed supplementary file, which contains a pdf file.
> > Please note that this supplemental PDF file does not show references or citations, and is only there to help with tracking
> > of updated notation. The new notation is in blue, while the old notation is in red.
> >
> > Respectfully yours.

---

### Decision · Action_Editor_YEvV · 2024-11-29

**Recommendation:** Accept as is

**Comment:**

After the rebuttal, all reviewers backed acceptance of the paper.  There was clear agreement that the updates provided have addressed the key concerns originally raised.  My own assessments did not bring up any reason to go against this unanimous conclusion, and I therefore recommend acceptance of the paper.   As there are no major outstanding issues, I recommend that the paper is accepted "as is".

**Audience:**

All reviewers and myself are in agreement that there is a clear audience who will be interested in the work.

**Claims And Evidence:**

Initially concerns were raised by reviewers about the empirical comparisons in the paper, e.g. to normalising flow methods.  However, significant new comparisons (both in terms of new baselines and new experiments themselves) were provided during the rebuttal and all reviewers are now happy with the empirical assessment provided by the paper.

Other issues were also raised around the clarity of the paper (in particular regarding the notation) and the discussion of the results.  Again though, these were well addressed during the rebuttal with no remaining objections in the final recommendations of the reviewers.

---

> ### Author Response · Authors · 2024-12-24
> **Submission of the camera-ready version**
>
> Dear AE,
>
> We appreciate your time handling our manuscript. We are grateful to you and the reviewers.
> We submitted the camera-ready version of our manuscript.
>
> Respectfully yours.